Applied and Environmental Science

# Geothermal Gases Shape the Microbial Community of the Volcanic Soil of Pantelleria, Italy

Nunzia Picone,[a] Carmen Hogendoorn,[a] Geert Cremers,[a] Lianna Poghosyan,[a] Arjan Pol,[a] Theo A. van Alen,[a] Antonina L. Gagliano,[b] Walter D'Alessandro,[b] Paola Quatrini,[c] Mike S. M. Jetten,[a] Huub J. M. Op den Camp,[a] Tom Berben[a]

[a]Department of Microbiology, IWWR, Radboud University, Nijmegen, Netherlands
[b]Istituto Nazionale di Geofisica e Vulcanologia, Sezione di Palermo, Palermo, Italy
[c]Department of Biological, Chemical and Pharmaceutical Sciences and Technologies (STEBICEF), University of Palermo, Palermo, Italy

Nunzia Picone and Carmen Hogendoorn contributed equally to this work. Author order was determined by drawing straws.

**ABSTRACT** Volcanic and geothermal environments are characterized by low pH, high temperatures, and gas emissions consisting of mainly $CO_2$ and varied $CH_4$, $H_2S$, and $H_2$ contents which allow the formation of chemolithoautotrophic microbial communities. To determine the link between the emitted gases and the microbial community composition, geochemical and metagenomic analysis were performed. Soil samples of the geothermic region Favara Grande (Pantelleria, Italy) were taken at various depths (1 to 50 cm). Analysis of the gas composition revealed that $CH_4$ and $H_2$ have the potential to serve as the driving forces for the microbial community. Our metagenomic analysis revealed a high relative abundance of *Bacteria* in the top layer (1 to 10 cm), but the relative abundance of *Archaea* increased with depth from 32% to 70%. In particular, a putative hydrogenotrophic methanogenic archaeon, related to *Methanocella conradii*, appeared to have a high relative abundance (63%) in deeper layers. A variety of [NiFe]-hydrogenase genes were detected, showing that $H_2$ was an important electron donor for microaerobic microorganisms in the upper layers. Furthermore, the bacterial population included verrucomicrobial and proteobacterial methanotrophs, the former showing an up to 7.8 times higher relative abundance. Analysis of the metabolic potential of this microbial community showed a clear capacity to oxidize $CH_4$ aerobically, as several genes for distinct particulate methane monooxygenases and lanthanide-dependent methanol dehydrogenases (XoxF-type) were retrieved. Analysis of the $CO_2$ fixation pathways showed the presence of the Calvin-Benson-Bassham cycle, the Wood-Ljungdahl pathway, and the (reverse) tricarboxylic acid (TCA) cycle, the latter being the most represented carbon fixation pathway. This study indicates that the methane emissions in the Favara Grande might be a combination of geothermal activity and biological processes and further provides insights into the diversity of the microbial population thriving on $CH_4$ and $H_2$.

**IMPORTANCE** The Favara Grande nature reserve on the volcanic island of Pantelleria (Italy) is known for its geothermal gas emissions and high soil temperatures. These volcanic soil ecosystems represent "hot spots" of greenhouse gas emissions. The unique community might be shaped by the hostile conditions in the ecosystem, and it is involved in the cycling of elements such as carbon, hydrogen, sulfur, and nitrogen. Our metagenome study revealed that most of the microorganisms in this extreme environment are only distantly related to cultivated bacteria. The results obtained profoundly increased the understanding of these natural hot spots of greenhouse gas production/degradation and will help to enrich and isolate the microbial key players. After isolation, it will become possible to unravel the molecular

Address correspondence to Huub J. M. Op den Camp, h.opdencamp@science.ru.nl.

mechanisms by which they adapt to extreme (thermo/acidophilic) conditions, and this may lead to new green enzymatic catalysts and technologies for industry.

**KEYWORDS** metagenomics, geothermal, methane, hydrogen, methanotroph, methanogenesis

Terrestrial volcanic/geothermal areas are extreme environments characterized by high temperatures, a low pH, and high gas emissions, of which the greenhouse gas carbon dioxide ($CO_2$) is dominant (1). Additionally, several reduced gases are emitted in variable concentrations, including hydrogen sulfide ($H_2S$), hydrogen ($H_2$), carbon monoxide (CO), ammonia ($NH_3$), and the potent greenhouse gas methane ($CH_4$) (2, 3). These gases are the driving force for the formation of a predominantly chemolithoautotrophic microbial community. Metagenomics analysis of the microbial community in hot springs with neutral pH in Yellowstone National Park shows an enrichment in genes involved in the metabolisms of these reduced volcanic gases (4, 5). Temperature, acidity, and soil characteristics further define the composition of microbial populations (6–8).

The island of Pantelleria represents the emerged part of an active volcanic system derived from the last eruption that occurred in 1891, 5 km northwest of its coast (9). It is the type locality of peralkaline rhyolitic rocks (pantellerites) (10). The island hosts a high-enthalpy geothermal field with a temperature of 250°C at a 1.5- to 2-km depth and whose surface expression is the fumarolic field of Favara Grande (11). With respect to other geothermal systems in the Mediterranean area, the released gases are particularly enriched in $H_2$ and $CH_4$ and poor in $H_2S$ (12). Previous work described the gas fluxes at two different sites, namely, FAV1 and FAV2, in the main gas-emitting area Favara Grande and provided indications for microbial $CH_4$ and $H_2$ oxidation within these soils (3, 6). In particular, 16S rRNA gene amplicon sequencing showed the presence of a diverse methanotrophic community, including *Methylocaldum*, *Methylobacter*, and *Beijerinckia* in FAV2 (6). Although only approximately 10 m apart, FAV1 did not contain many methanotrophs but instead was dominated by thermoacidophilic chemolithotrophs. In both sites, *Proteobacteria* represented the most abundant bacterial phylum, and some archaeal reads were detected (6).

*Proteobacteria* are often found in volcanic/geothermal ecosystems (6, 13–15), and this phylum includes methanotrophic and methylotrophic bacteria. The phylum *Verrucomicrobia* was shown to contain methanotrophs as well, with cultivated methanotrophic *Verrucomicrobia* being able to grow at low pH and, in case of the genus *Methylacidiphilum*, also at high temperatures. The cultivated strains are isolated from mud volcanoes and geothermal soils all over the world (16–20). These *Verrucomicrobia* are often not detected in geothermal soils (6, 15), as their *pmoA* and 16S rRNA gene sequences are not very well amplified by commonly used PCR primers (21). This may be one of the reasons why these methanotrophs were absent in previously acquired amplicon sequencing data of the Solfatara and Pantelleria regions (6, 15, 16). However, Gagliano et al. (22) were able to retrieve verrucomicrobial *pmoA* sequences after designing specific primers.

Besides $CH_4$, $H_2$ is an abundant gas in volcanic and geothermal ecosystems that can be used by members of the microbial community as an electron donor (23, 24). The low $O_2$ concentrations within these soils facilitate microaerobic $H_2$ oxidation using oxygen-sensitive hydrogenases. Localized spots without any $O_2$ could give rise to anaerobic $H_2$ consumption, for example, by autotrophic sulfate-reducing microorganisms (25) or hydrogenotrophic methanogens (26). Methanogens are often detected in volcanic/geothermal environments, especially in the deeper anaerobic layers of the soil (27).

Oxidation of $H_2$ is catalyzed by hydrogenase enzymes, which can be grouped as [NiFe]-, [FeFe]-, or [Fe]-hydrogenases depending on the metal ion(s) in the active site (28). Greening et al. (29) categorized the hydrogenases in eight groups based on amino acid sequence, phylogeny, metal-binding motifs, predicted genetic organization, and reported biochemical characteristics. The groups and subgroups show clear differences

**TABLE 1** Temperature, pH, chemical gas composition of the Favara Grande sampling site

| Sample site | Depth (cm) | Temp. (°C) | pH | Composition (ppm)[a] | | | | | | |
|---|---|---|---|---|---|---|---|---|---|---|
| | | | | He | $H_2$ | $O_2$ | $N_2$ | $CH_4$ | $CO_2$ | $H_2S$ |
| FAV1 | 11 | 61.1 | 3 | 16 | 32,800 | 2,700 | 17,600 | 38,500 | 891,400 | 700 |
| | 20 | 74.1 | 3.5 | 20 | 45,200 | 5,600 | 26,200 | 40,000 | 860,900 | 300 |
| | 30 | 87.4 | 4 | 28 | 69,900 | 4,800 | 22,100 | 42,100 | 844,500 | 200 |
| | 50 | 98.9 | 4 | 28 | 77,100 | 5,700 | 25,200 | 39,900 | 834,100 | 200 |
| FAV2 | 11 | 60.2 | 4 | <5 | 125 | 192,500 | 739,400 | 1,000 | 22,000 | <50 |
| | 20 | 67.3 | 4.5 | 8 | 8,400 | 107,000 | 411,100 | 18,000 | 426,700 | <50 |
| | 30 | 77.7 | 4 | 13 | 25,000 | 4,100 | 17,000 | 38,500 | 893,200 | <50 |
| | 50 | 91.7 | 4 | 21 | 48,900 | 3,300 | 13,800 | 40,300 | 874,100 | <50 |

[a]The analytical errors were less than ±10% for He and less than ±5% for the remaining gases.

in several properties, such as $O_2$ tolerance and direction of the reaction, i.e., $H_2$ consuming, $H_2$ evolving, or bidirectional. Furthermore, some hydrogenases are classified as sensory hydrogenases that control the expression of respiratory hydrogenases (29, 30). The capacity for $H_2$ oxidation is present in many phyla among both *Bacteria* and *Archaea* (29), including verrucomicrobial methanotrophs (31–34).

Geothermal ecosystems are low in organic carbon, but $CO_2$ is highly available and can be assimilated into biomass by chemolithoautotrophic microorganisms. In total, six $CO_2$ fixation pathways have been discovered, namely, the Calvin-Benson-Bassham (CBB) cycle, the 3-hydroxypropionate cycle, the 3-hydroxypropionate-4-hydroxybutyrate cycle, the reductive tricarboxylic acid (TCA) cycle, the Wood-Ljungdahl pathway, and the dicarboxylate/4-hydroxybutyrate cycle (35, 36). Fixing inorganic carbon requires energy and oxidation of the reduced geothermal gases can provide this energy.

It remains difficult to link 16S rRNA gene amplicon results to the metabolic potential of the microorganisms identified. Therefore, studying the presence of functional genes is a good approach to reveal the potential for microbe-mediated geothermal gas oxidation, including $CH_4$ and $H_2$ and $CO_2$ fixation. In this study, we used metagenomic sequencing rather than PCR-driven methods to reduce primer bias and to give a better insight into the microbial diversity and the metabolic potential of this community.

The aim of our research was to determine the important gaseous electron donors that could support the chemolitho(auto)trophic microbial community on the volcanic island of Pantelleria and directly link this to the microbial community present. This microbial population was investigated using metagenome sequencing and the detection of key microbial players involved in the oxidation of $CH_4$ and $H_2$, including some that might have been overlooked in previous analyses. Furthermore, we used the metagenomics data to predict metabolic pathways involved in the oxidation of gases and link these to gas measurements and microbial community composition.

## RESULTS AND DISCUSSION

**Geochemical characteristics of the sampling site.** The Favara Grande is the main gas-emitting area of Pantelleria Island, Italy, characterized by numerous gaseous manifestations due to geothermal activity (3, 22). A geochemical analysis was performed to detect which reduced gases were available for use by the microbial community. For this purpose, soil and gas samples were collected in June 2017 at the sites FAV1 and FAV2 (22). Although these sites were only approximately 10 m apart, they strongly differed in gas composition, pH, and temperature gradient (Table 1). These differences, even at sampling spots very close to each other, are due to structural discontinuities, e.g., faults or cracks, allowing preferential routes for the upflow of hydrothermal gases (37). Cracks present in actively degassing geothermal soils, like those of Pantelleria, are of great importance since gas is often pressure-driven, contrarily to nongeothermal soil where gas movements are generally only driven by concentration gradients. Gas flow is, therefore, often focused in small areas, sometimes becoming open vents (fumaroles), where hydrothermal gases are directly released into the atmosphere. The gases themselves interact and modify the soil, and the most effective agents are temperature and

pH. Temperature, pH, and soil properties depend on the energy of the gases escaping from the underground. Kinetic/thermal energy allows them to flow up and rules the gas-water-rock interactions. Due to these interactions, during their ascent to the surface, the volcanic/geothermal gases are depleted in some species (such as $SO_2$) and enriched in others (such as $CH_4$ and $H_2S$) (38). With their residual energy, these gases reach the surface, and interacting with altered soil horizons may decrease their permeability (37). When the soil is highly altered, as at FAV1 and in the deeper layer of FAV2 (6, 39), the diffusion of the air into the soil is slow while the hydrothermal gas upflow is faster, hampering the air dilution of the hydrothermal gases and the soil aeration.

At 20-, 30-, and 50-cm depths, the temperature of FAV1 was higher than that of FAV2. The pH in FAV2 was between 4 and 4.5 throughout the core, whereas the pH in FAV1 decreased toward the surface, reaching a pH of 3 at the 1- to 10-cm depth. This coincides with the higher $H_2S$ concentrations at FAV1.

In the deeper soil layer, the hydrothermal gases contained large amounts of $CO_2$, $H_2$, and $CH_4$. Near the surface, these gases mixed with air, especially in FAV2, where the $O_2$ and $N_2$ concentrations were much higher at the 1- to 10-cm depth and decreased toward deeper layers. $H_2$ and $CH_4$ fractions, instead, showed an opposite trend, and their concentrations decreased toward the surface. This counter gradient could enable biological activity. In FAV1, there seemed to be less mixing with air, probably due to different soil permeability and weathering (6) that influences the hydrothermal flux locally. The $H_2$ fraction also decreased toward the surface in FAV1, but $CH_4$ remained constant over the whole depth, indicating only $H_2$ consumption at this site (6). Other minor gases included (in variable concentrations) hydrogen sulfide ($H_2S$, only measured in FAV1), carbon monoxide (CO), and helium (He).

Compared to previous data (6, 22), the concentrations of $CH_4$, $H_2$, and $CO_2$ were overall similar. $N_2$ and $O_2$ concentrations, instead, appeared to be lower in our measurements, especially in the deeper layers of the soil. It needs to be noted, though, that the depths where the sampling took place slightly varied, except for 50 cm (6, 22). For this reason, it remains difficult to directly compare other physical parameters, such as temperature and pH. At 50 cm, the temperature measured during our sampling campaign was 3.8°C lower than previous recordings in FAV1 and 19.9°C lower in FAV2. pH, instead, was measured only at 0 to 3 cm, and pH values showed comparable values in FAV1 but higher values than in FAV2 (pH 5.8 versus pH 4).

The soil gas composition is governed by three main processes, (i) mixing between two end members (uprising hydrothermal gases and atmospheric air diffusing within the soil), (ii) dissolution in condensing water vapor, and (iii) biological consumption or production. The first process is well evidenced in the FAV2 profile, where the deepest samples show a composition very close to the fumarolic gases of the Favara Grande area (3), while the shallowest sample has a composition close to that of the atmospheric air. Typical hydrothermal gases (He, $H_2$, $CH_4$, and $CO_2$) decrease their content going toward the soil surface, while atmospheric components ($O_2$ and $N_2$) show an opposite trend. In the case of FAV1, the hydrothermal component prevails along the whole profile. In this case, higher hydrothermal gas flux and/or lower soil permeability prevents a significant diffusion of the atmospheric component within the soil. The second process is important only for highly soluble gases such as $CO_2$ and $H_2S$. In the present case, $CO_2$ was scarcely affected, and this may be the reason why $H_2S$ was below the detection limit in FAV2. As also evidenced previously (6), the decrease of $H_2$ and $CH_4$ toward the soil surface does not perfectly follow the air dilution process, highlighting the effects of the third, biological, process. Furthermore, the gaseous emissions seem to vary over time and biotic factors, and the microbial consumption of geothermal gases could influence the geothermal gas emissions (2, 40, 41). To determine whether the potential for microbial consumption of these gases existed, we investigated the local microbial community.

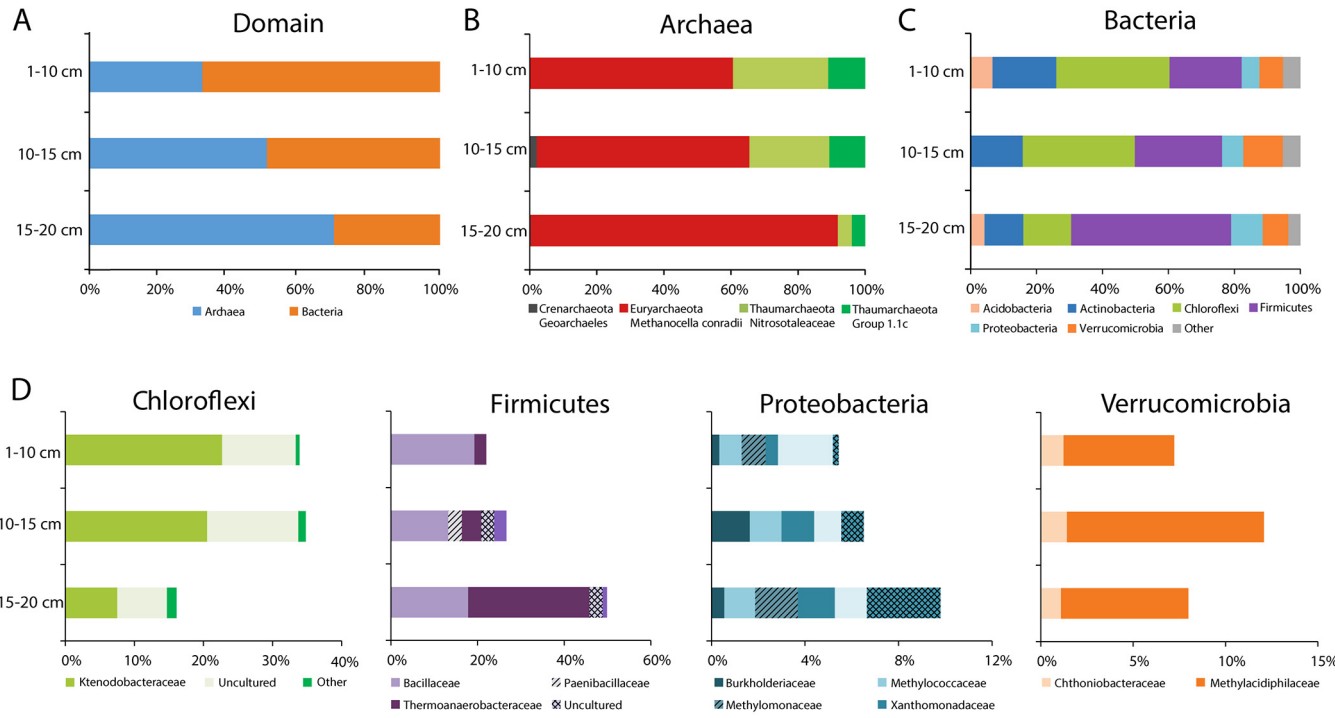

**FIG 1** Community composition of the soil of Pantelleria Island at FAV2 based on 16S rRNA gene sequences retrieved from the metagenome. (A) Relative abundance at the domain level. (B and C) Relative abundances of the different phyla and families in the domains *Archaea* and *Bacteria*. (D) Distributions of the most abundant bacterial families in the phyla *Chloroflexi*, *Firmicutes*, *Proteobacteria*, and *Verrucomicrobia*. Taxonomic groups with relative abundance of <5% are classified as "other." A complete list of the relative abundance of each group, calculated for the different DNA extraction methods, can be found in Table S1 in the supplemental material.

**Community analysis.** To study the microbial community of Favara Grande, soil core samples from FAV1 and FAV2 were taken and subsampled at different depths. An attempt to extract DNA from 1 to 10 cm, 10 to 15 cm, and 15 to 20 cm of both sites was made, but high-quality DNA was only retrieved from FAV2 samples. To reduce biases due to the extraction method (42), two different protocols were used. Illumina MiSeq sequencing of the DNA isolated with cetyltrimethylammonium bromide (CTAB) and a Fast DNA Spin kit resulted in a total of 112,613,756 reads evenly distributed over the sampling depths (for details, see Materials and Methods). From these data sets, reads mapping to 16S rRNA genes were extracted and assembled. In total, 62 full-length 16S rRNA genes were retrieved, and their relative abundance was calculated (Fig. 1).

The soil microbial community of FAV2 contained 68% of *Bacteria* at the 1- to 10-cm depth that were gradually replaced by *Archaea* (70%) in the deeper layer (Fig. 1A). Nearly all archaeal 16S rRNA gene reads mapped to a single species of a hydrogenotrophic methanogen, related to *Methanocella conradii* (99% 16S rRNA gene identity) (Fig. 1B) (43). Such a high relative abundance of a methanogen was unexpected and not observed before, as previous gas analyses indicated that most if not all $CH_4$ was of thermogenic and abiotic origin (3). Furthermore, geochemical characteristics of the soil differed quite substantially from the growth conditions of the cultivated *M. conradii* HZ254. Although strain HZ254 has an optimum growth temperature of 55°C, which is slightly below the temperature measured in the soil (Table 1), it only grows at a pH between 6.4 and 7.2 (43), which is much higher than the pH that was measured at the FAV2 site. Furthermore, the $O_2$ concentration in the deeper layers was relatively high for an anaerobic methanogenic microorganism to thrive (Table 1). Methanogens are strictly anaerobic and exposure to $O_2$, even briefly, can reduce methanogenic activity greatly (44, 45). Despite this oxygen sensitivity, methanogenesis has been widely observed in environments with low oxygen levels, which is known as the "methane paradox," and

it is of great environmental significance as it may account for 40% to 90% of emitted $CH_4$ (46–48). Aerobic methanogenesis is not performed by *Archaea*, but aquatic and terrestrial cyanobacteria can produce minor amounts of $CH_4$, under oxic conditions, presumably from methylphosphonate (49–51). For *Methanocella* to be active, it is reasonable to assume that anaerobic conditions must be present in the deeper layers of FAV2. One hypothesis is that oxygen depletion by aerobic members of the microbial community in biofilms or microaggregates may create conditions for anaerobic methanogenic *Archaea* to thrive.

The remainder of the archaeal community consisted of *Thaumarchaeota*. "*Ca*. Nitrosotaleaceae" was the most abundant family within this phylum, which includes acidophilic autotrophic ammonia-oxidizing archaea, with a high affinity for ammonia (52–54). We were not able to detect ammonia in the gases emitted in FAV2 by passing gas samples through Milli-Q water at the sampling site and determining the amount of $NH_4^+$ by the sensitive *o*-phthalaldehyde method. However, we were able to measure 1.7 $\mu$M nitrite at site FAV2. In FAV1, instead, we detected up to 74 $\mu$M ammonium and 1 $\mu$M nitrite.

The bacterial community was more diverse than the archaeal one: the dominant phyla were *Actinobacteria*, *Chloroflexi*, *Firmicutes*, and *Verrucomicrobia*. *Chloroflexi* were most abundant at 1 to 10 cm and at 10 to 15 cm and showed a decreasing relative abundance with increasing depth. *Firmicutes* were dominant at 15 to 20 cm (Fig. 1B), and their relative abundance increased in the deeper layers (Fig. 1C). Within the *Firmicutes*, the family *Bacillaceae* was equally present across all depths, but at 15 to 20 cm, the relative abundance of the family *Thermoanaerobacteraceae* increased. These thermophiles conserve energy through fermentation; however, some can grow chemolithoautotrophically (55, 56). A Simpson index (alpha diversity) plot of the three horizons split up between *Bacteria* and *Archaea* can be found in Fig. S1 in the supplemental material.

The microbial community composition differed from the results previously obtained using 16S rRNA gene amplicon sequencing (6). In that study, *Proteobacteria* were the dominant phylum with a relative abundance of 48% (6). In the 16S rRNA gene information extracted from our metagenomic data, *Chloroflexi* and *Firmicutes* dominated the bacterial population instead, and *Proteobacteria* showed a relative abundance of less than 10%. Furthermore, *Verrucomicrobia* were not detected in the previous amplicon data (6), but verrucomicrobial *pmoA* sequences were retrieved before from the same site using specifically designed primers (22). In our metagenome study, this phylum accounted for 7% to 12% of the bacterial community. This discrepancy, beside the change that might naturally happen over time, can also be attributed to the different techniques used for analysis, in particular, to biases introduced in the amplification step of amplicon sequencing and the DNA extraction method (21).

*Proteobacteria* and *Verrucomicrobia* are of special interest, since these phyla contain aerobic methanotrophs. Within the phylum *Verrucomicrobia*, the family *Methylacidiphilaceae* contains cultivated acidophilic methane oxidizers (19, 20, 57), and members of this family were detected within the FAV2 geothermal soils (Fig. 1D). In addition, gammaproteobacterial methanotrophic sequences were retrieved from the FAV2 soil metagenome. They belonged to the families *Methylomonaceae* and *Methylococcaceae* (Fig. 1D), with a clear difference in relative abundance over depth: the amount of *Methylococcaceae* remained more or less constant, but *Methylomonaceae* were detected at 1 to 10 cm and in the 15- to 20-cm layer but not in the middle layer. The verrucomicrobial methanotrophs of the family *Methylacidiphilaceae* showed a higher relative abundance than the gammaproteobacterial methanotrophs at all depths. These results indicate that verrucomicrobial methanotrophs might contribute more to methane oxidation than methanotrophs belonging to the *Gammaproteobacteria* in the hot acidic soil of Pantelleria. The gammaproteobacterial methanotrophs described so far are less acidophilic than *Verrucomicrobia*, and no isolate was shown to grow below pH 4.2 (58, 59).

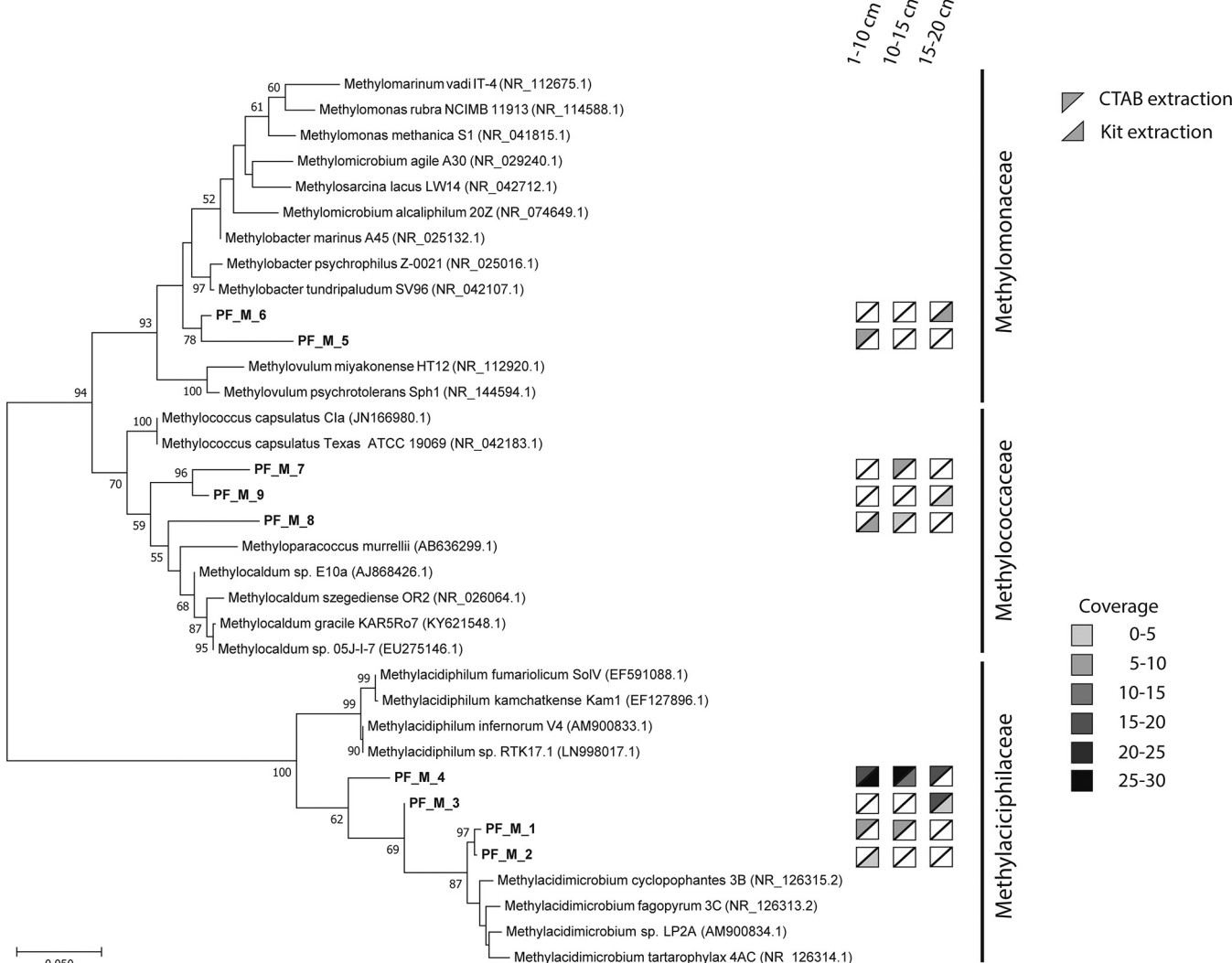

**FIG 2** Maximum likelihood phylogenetic tree of 16S rRNA genes of methanotrophs at the Favara Grande soil based on 1,000 bootstrap replicates (values of >50 are shown). The gray triangles indicate the DNA extraction method. The shading gradient indicates the coverage of the sequences at different depths (white, not detected). The tree was rooted using the 16S rRNA sequences of *Nitrospira moscoviensis*, but this branch was removed for clarification.

Examining the 16S rRNA gene sequences at the species level showed a clear difference in the distributions of these verrucomicrobial methanotrophs according to depth (Fig. 2). Furthermore, the effect of DNA extraction bias was clearly observed. The most abundant *Verrucomicrobia* (sequences PF_M_3 and _4) were detected with both DNA extraction methods, but the rarer methanotrophs (sequences PF_M_1 and _2) were only detected with one of the DNA extraction methods (Fig. 2). Furthermore, these results showed that the composition of the methanotrophic community changed with depth likely due to changing conditions, such as temperature, geothermal gas composition, and $O_2$ concentration. For example, PF_M_3 was only detected at 15 to 20 cm, whereas PF_M_1 and PF_M_2 DNA was found in the upper layers of the soil. To investigate the capacity of the microbial players to utilize the geothermal gases, the key genes for main metabolic pathways were analyzed in more detail.

**Metabolic potential of the soil microorganisms.** $H_2$ and $CH_4$ are the main reduced gases that can be used by microorganisms as an energy source in the core here studied. To determine whether the microbial community is capable of oxidizing these gases, we mined our metagenome for functional genes by hidden Markov model (HMM) profiling (60). From this analysis, the metabolic potential for carbon and hydrogen cycling was

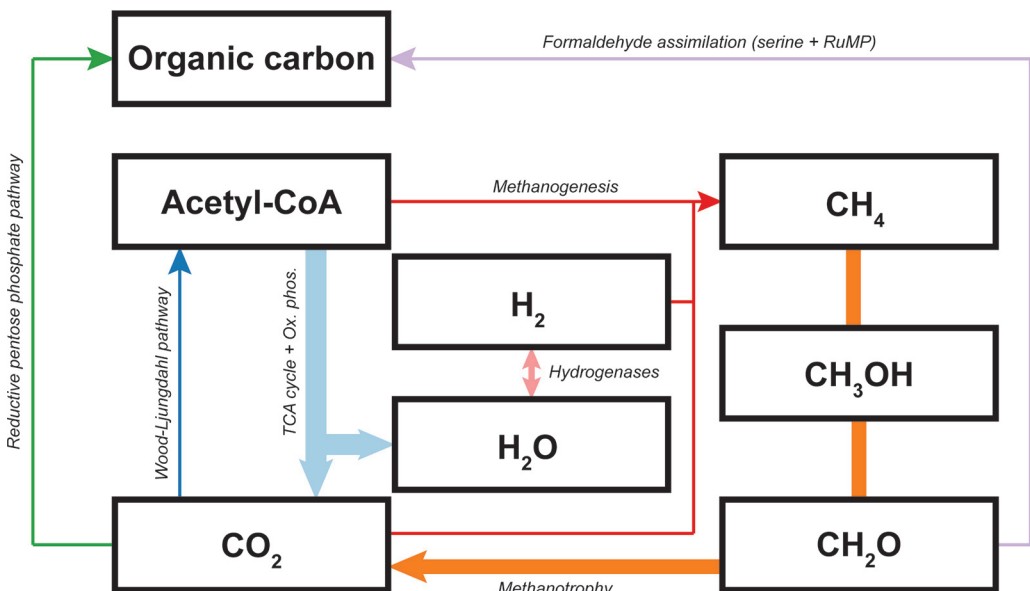

**FIG 3** Carbon and hydrogen cycling capacity in the microbial community. Arrow thickness represents the percentage of genes involved in that pathway. Methanotrophy, 13.1%; methanogenesis, 1%; hydrogen oxidation, 3.5%; serine pathway, 2.5%; ribulose monophosphate (RuMP), 0.3%; reductive pentose phosphate pathway (also called Calvin-Benson-Bassham cycle), 1%; Wood-Ljungdahl pathway, 1%; TCA cycle, 9.1%; oxidative phosphorylation, 18.9%.

determined (Fig. 3), together with different inorganic carbon fixation pathways and some genes encoding enzymes important in nitrogen and sulfur cycles (Fig. 4A). The next paragraphs provide more details on the metabolic potential analyses.

**Methanotrophy and methylotrophy.** The emitted $CH_4$ can potentially be oxidized by soil microorganisms, as 13.1% of the retrieved genes encode proteins involved in the $CH_4$ oxidation pathway (Fig. 3). These proteins include the particulate methane mono-oxygenase (pMMO), the methanol dehydrogenase (MDH), formaldehyde dehydrogenase, and formate dehydrogenase. Sequences encoding the soluble MMO (sMMO) were not detected in our data set. The pMMO enzyme is highly similar to the ammonia monooxygenase (AMO) enzyme found in ammonia-oxidizing *Bacteria* and *Archaea* (52, 61); therefore, the retrieved *pmoA* sequences need to be phylogenetically analyzed to distinguish between *pmoA* and *amoA* genes. None of seven *pmoA-amoA* sequences clustered with known *amoA* genes, indicating that there are no bacterial/archaeal ammonia monooxygenases detected within this metagenome (Fig. 5) despite 16S rRNA gene retrieval of potential ammonia-oxidizing *Thaumarchaeota* (Fig. 1). The phylogenetic analysis of the retrieved *pmoA* sequences showed that three of them clustered within the *Gammaproteobacteria* and two within the phylum *Verrucomicrobia*. Sequence ms_fav_M_1 clustered within the *pmoA* group but was only distantly related to known *pmoA* sequences, making it impossible to assign a meaningful taxonomic level to it (Fig. 5). Analysis of the *pmoA* and the 16S rRNA gene sequences showed that methanotrophy in the geothermal soils of Pantelleria Island was performed by members of the phyla *Verrucomicrobia* and *Gammaproteobacteria* but not *Alphaproteobacteria*. These results are in agreement with the microbial community analysis (Fig. 1).

The second step in the methane oxidation pathway is the conversion of methanol to formaldehyde catalyzed by the enzyme methanol dehydrogenase (MDH). For methylotrophs, it is the first step in their metabolism. In the metagenome, 104 MDH sequences were identified, 58 of which were small fragments and therefore discarded for further analysis. To classify the retrieved MDH, the full-length sequences were integrated into the phylogenetic tree reported by Keltjens et al. (62), showing that all retrieved MDH sequences were pyrroloquinoline quinone (PQQ)-dependent XoxF-type MDH and that no calcium-dependent MDH sequences were found (see Fig. S2). Lanthanide-dependent MDHs have better catalytic characteristics than their calcium-

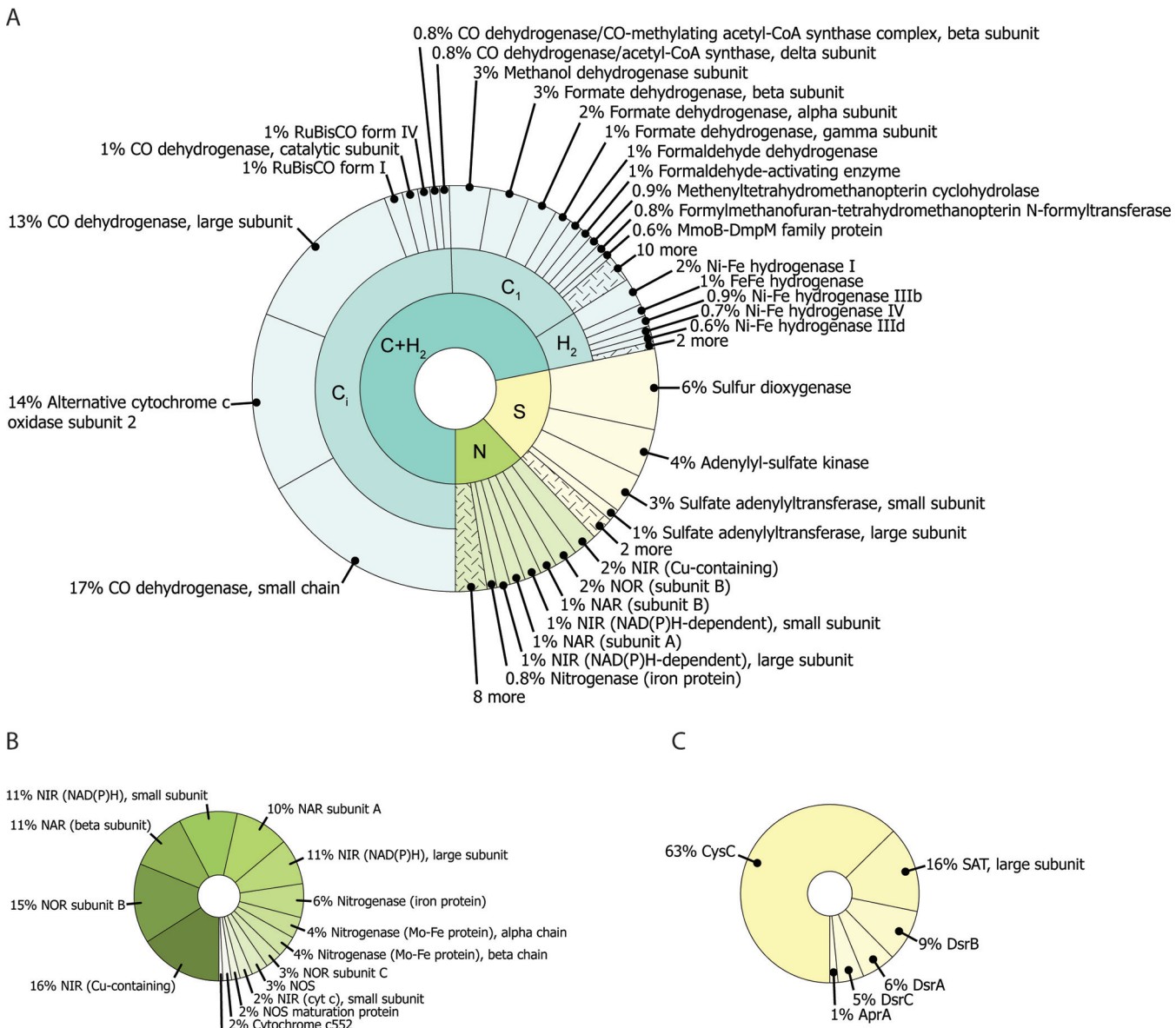

**FIG 4** Biochemical capacity of the microbial community. (A) The key genes for carbon, hydrogen, sulfur, and nitrogen metabolism are reported as percentage of annotated genes. (B) Nitrogen metabolism. Genes for $N_2$ fixation and denitrification are the most abundant. (C) Sulfur metabolism.

dependent counterparts (63, 64). Typically, the bioavailability of lanthanides is poor due to their low solubility; however, the solubility of these elements increases at low pH. Since there were only lanthanide-dependent MDHs detected in the metagenomic data set, this indicates lanthanides are sufficiently available for the microbial community and there is no niche for the calcium-dependent methanol dehydrogenases.

The lanthanide-dependent MDHs can be classified into five groups (62). The detected MDH sequences were distributed across XoxF1, XoxF2, XoxF3, and XoxF5 and a novel group named XoxF6. The majority of the sequences grouped with an XoxF3-type MDH. This protein is found in a large variety of bacterial species belonging mainly to the phyla *Proteobacteria* or *Acidobacteria*, including the genera *Methylobacter*, *Bradyrhizobium*, *Burkholderia*, "*Candidatus* Solibacter," etc. (Fig. S2). They are $N_2$ fixers (*Bradyrhizobium* and *Burkholderia*), methanotrophs, or methylotrophs found in different environments, such as hot springs and North Sea sediments (65, 66). Some of these species have multiple copies of MDH, suggesting that XoxF3 could be alternatively transcribed under specific conditions, as found for other enzymes (31, 67).

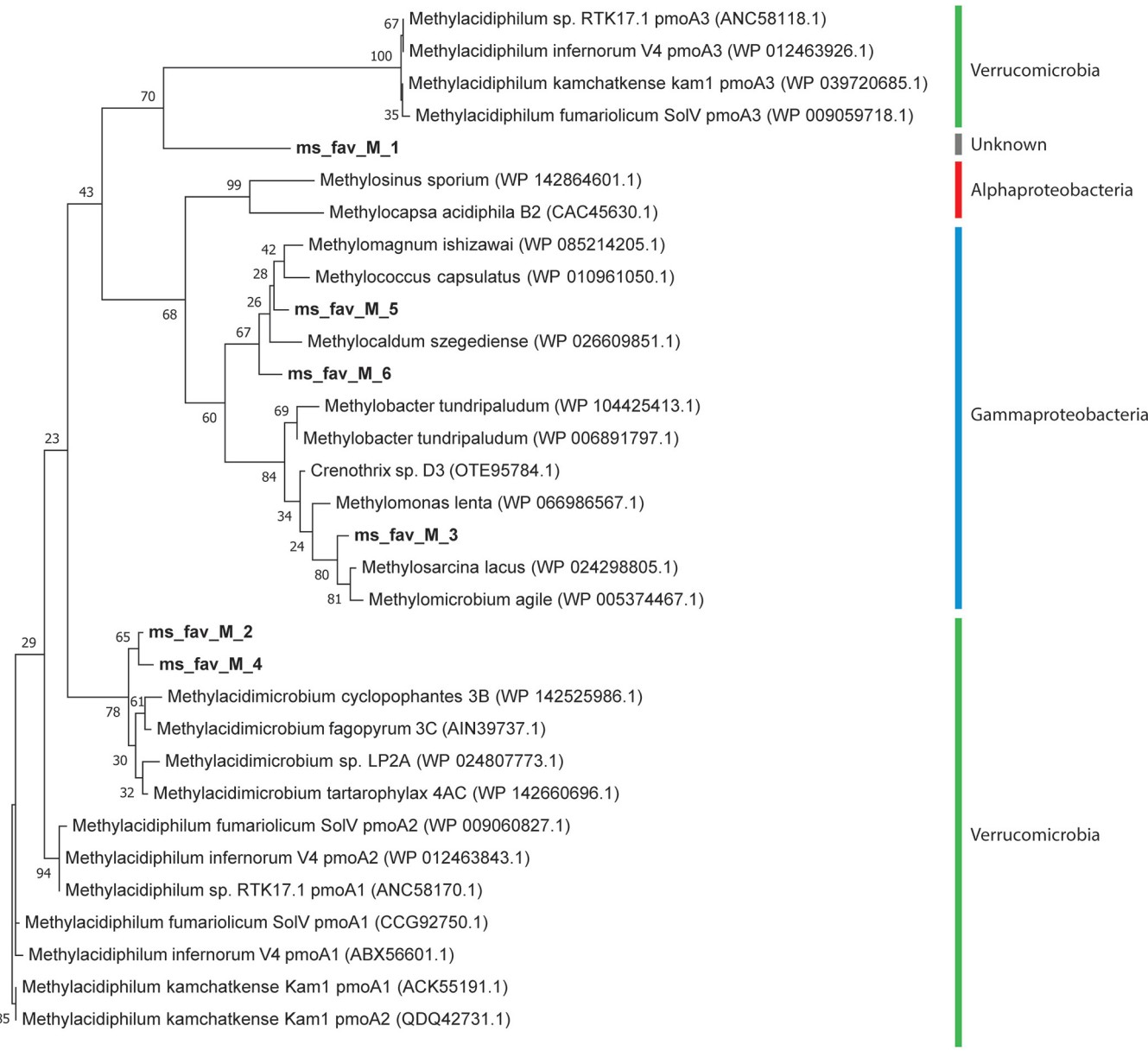

**FIG 5** Maximum likelihood phylogenetic tree of *pmoA* and *amoA-pmoA*-like genes retrieved from the metagenome based on 1,000 bootstrap replicates. The tree was rooted with the *amoA* gene of *Nitrosopumilus* sp. (WP_016939485.1), but this branch was removed for clarification. The colored bars indicate the phylum or order that the sequence is affiliated with.

XoxF1 and XoxF2 were also found in the metagenome (Table 2). XoxF2 genes are mainly harbored by bacteria of the phylum *Verrucomicrobia*. XoxF1 is found in *Verrucomicrobia*, anaerobic NC10 bacteria, and *Proteobacteria* (68). Additionally, we detected the presence of a novel group of XoxF MDHs, which has been named "XoxF6." XoxF6, together with XoxF1, XoxF3, and XoxF5, is found also in *Alphaproteobacteria* (Fig. S2). This result indicated that methylotrophic *Alphaproteobacteria* may also play an important role in the carbon cycle of the geothermal soils of Pantelleria, as was observed in marine seeps exposed to high abiotic $CH_4$ release (69).

**$H_2$ metabolism.** To investigate the metabolic potential for $H_2$ metabolism, the metagenome was mined for hydrogenase genes. In total, 436 hydrogenase genes were

**TABLE 2** XoxF type methanol dehydrogenases retrieved from the metagenome

| XoxF type | No. of genes |
|-----------|--------------|
| XoxF1 | 2 |
| XoxF2 | 3 |
| XoxF3 | 35 |
| XoxF5 | 5 |
| XoxF6 | 1 |

retrieved, which accounted for 3.5% of all annotated genes (Fig. 3). Most of the hydrogenase sequences encode [NiFe]-hydrogenases, and the rest belong to the [FeFe]-hydrogenases. No [Fe]-hydrogenase-encoding genes were found (Table 3).

Group 4 ([NiFe]-evolving hydrogenases) showed the highest relative abundance (Table 3). These hydrogenases have been known for their role in fermentation, but recent studies showed that they can also function in respiration in anaerobic *Bacteria* and *Archaea* (70). During anaerobic growth on CO or formate, they produce $H_2$ while oxidizing reduced ferredoxin (71). Interestingly, these evolving [NiFe]-hydrogenases are typically not abundant in geothermal soils, but they are mostly found in permafrost. Bidirectional [NiFe]-hydrogenases, instead, are usually retrieved from hot spring metagenomes (30).

Although many different [FeFe]-hydrogenases were detected in the metagenome (Table 3), the relative abundance of this type of hydrogenase was low (Table 3). This was in contrast to the [NiFe]-anaerobic uptake hydrogenases which formed the second-most-abundant group, after the [NiFe]-evolving hydrogenases (Table 3). [NiFe]-anaerobic uptake hydrogenases are involved in the anaerobic respiration of $H_2$ (30). However, it was recently shown that these anaerobic uptake hydrogenases can also play a role in aerobic $H_2$ oxidation under very low $O_2$ concentrations (31, 33). In the Favara Grande ecosystem, the $O_2$ concentrations that were measured in the top 20 cm of the soil (Table 1) exceeded the concentrations at which these anaerobic uptake [NiFe]-hydrogenases are known to be active. Keeping in mind that the presence of genomic DNA is not necessarily an indication of functional activity, these results, together with the relative high abundance of methanogens, might imply that there are spatial or temporal niches where $O_2$ is depleted and anaerobic metabolism can play a significant role.

Comparing the distribution of hydrogenases of the Favara Grande to those in other ecosystems shows that anaerobic uptake [NiFe]-hydrogenases appear to be present in deep oceans. Metagenomics studies have shown that bidirectional [NiFe]-hydrogenases are usually found in forest soil and permafrost, whereas aerobic uptake [NiFe]-hydrogenases showed high relative abundance in agricultural soils (29). There is no experimental evidence about what ecological factors drive the distribution of hydrogenases in a particular environment, but it was proposed that partial oxygen concentrations, pH, temperature, and metal ion availability are likely to play a role (29).

**Carbon fixation.** Alphaproteobacterial and gammaproteobacterial methanotrophs/methylotrophs assimilate carbon via organic compounds, such as formaldehyde. More

**TABLE 3** Numbers of hydrogenase genes and their relative abundance in geothermal soils of the Favara Grande

| Type of hydrogenase[a] | No. of genes | Relative abundance of genes (%) |
|------------------------|--------------|---------------------------------|
| [NiFe] Evolving | 192 | 65 |
| [NiFe] Anaerobic uptake | 10 | 20 |
| [FeFe] Hydrogenase | 187 | 7 |
| [NiFe] Bidirectional | 42 | 6 |
| [NiFe] Aerobic uptake | 8 | 2 |

[a]The [NiFe]-aerobic uptake hydrogenases only include subgroup 2a, [NiFe]-anaerobic uptake hydrogenases include subgroups 1b, 1c, 1f, and 1k, [NiFe]-bidirectional includes 3b and 3d, the [NiFe]-evolving hydrogenases include groups 4a, 4b, 4e, 4f, and 4g, and [FeFe]-hydrogenases include groups A, B, and C. The remaining subgroup, including groups 1a, 1d, 1e, 1g, 1h/5, 2b, 2c, 2d, 3a, 3c, 4c, and 4d and [Fe]-hydrogenases, were not identified within this metagenome.

specifically, the gammaproteobacterial methanotrophs/methylotrophs incorporate formaldehyde via the ribulose monophosphate pathway (RuMP), whereas alphaproteobacterial methanotrophs/methylotrophs use the serine pathway (72). These strategies for carbon assimilation seemed to be the most represented in the Pantelleria soil community (Fig. 3).

*Verrucomicrobia*, but also other methanotrophs, use the Calvin-Benson-Bassham cycle (or reductive pentose phosphate pathway) for $CO_2$ fixation (73–76). This pathway only accounted for a small part of the total number of genes (1%) (Fig. 3). The Calvin cycle is typical of photosynthetic organisms, including *Cyanobacteria* (35), but is also found in bacteria of the phyla *Verrucomicrobia* (74), *Proteobacteria* (77), *Actinobacteria* (78), and *Firmicutes* (79). All of these phyla were detected in the metagenome (Fig. 1).

The Wood-Ljungdahl (WL) pathway for $CO_2$ fixation was also detected (1%) (Fig. 3), and it is usually present in acetogenic and methanogenic microorganisms. The key enzyme, CO dehydrogenase, appeared to have a high relative abundance (Fig. 4A) in this volcanic ecosystem. CO dehydrogenases are enzyme complexes composed of different subunits (80); in this case, they accounted for a large part of the carbon metabolism, although the catalytic subunit represented only 1% of the genes compared to the large and small subunits that accounted for 13% and 17%, respectively. This discrepancy can be attributed to assembly limitations that might hamper the recovery of complete genes. The presence of the Wood-Ljungdahl pathway can be related to the abundance of *Archaea* in the soil, particularly, the methanogen *M. conradii* (Fig. 1B). The reverse TCA cycle can also generate acetyl coenzyme A (acetyl-CoA) starting from two molecules of $CO_2$, and it is found in *Archaea* and in anaerobic or microaerobic *Bacteria* (79, 81). Within the microbial population of Pantelleria, the sum of both the reductive and the oxidative pathways accounted for 9.1% of the total number of genes (Fig. 3), representing the second-most-utilized pathway for carbon fixation. Since both the Wood-Ljungdahl pathway and the reverse TCA cycle show a high relative abundance, anaerobic microorganisms are likely important primary producers. These anaerobic microorganisms might play an important role in the carbon cycle, together with methanotrophs that aerobically fix carbon.

**Nitrogen and sulfur metabolism.** The potential for an active sulfur and nitrogen metabolism was also detected. In particular, most of the retrieved nitrogen metabolism genes were involved in the denitrification process. Despite the presence of *Thaumarchaeota* (Fig. 1), a phylum that contains many ammonia oxidizers, no ammonia monooxygenase genes were retrieved (Fig. 4). This observation could be related to the geochemical characteristics of the soil of Pantelleria Island, as site FAV2 had a low concentration of ammonia (6, 18) and therefore is probably not much of a niche for ammonia oxidizers. Not all *Thaumarchaeota* are capable of ammonia oxidation, such as "*Candidatus* Caldiarchaeum subterraneum," and employ different metabolic strategies to survive (82). Additionally, dissimilatory ammonia oxidizers would have to compete for ammonia with organisms that assimilate it for biomass. Ammonia can be formed through the reduction of $N_2$ by the enzyme nitrogenase. This enzyme was detected and accounted for 0.8% of the total annotated genes (Fig. 4B). However, activity of this enzyme is restricted to microaerophilic environments (83) and most likely involved in $N_2$ fixation for biomass production.

The sulfur cycle in FAV2 is limited to a number of low-abundance genes. Sulfur dioxygenase was the most abundant of these, representing a sulfide detoxification system, whereas the other genes form a sulfate-reducing pathway, including sulfate adenylyl transferase, adenylyl-sulfate reductase, and adenylyl-sulfate kinase, and both assimilatory- and dissimilatory-type sulfite reductases showed a low abundance. Measured sulfide levels at FAV2 were <50 ppm, indicating that sulfur oxidation is likely not a major biogeochemical process. This is additionally supported by the fact that the pH of 4 is constant throughout the different soil layers, whereas soils in which sulfur metabolism is active are typically strongly acidified (complete sulfide oxidation leads to the formation of two protons from sulfuric acid), such as in Solfatara, near Naples, Italy

(84). The geochemical and 16S rRNA gene amplicon sequencing data of FAV1 supports sulfur cycling, including $H_2S$ emission and a decreasing pH (Table 1) (6).

**Conclusions.** The geothermal soil of Pantelleria shows high concentrations of $CH_4$ and $H_2$, which are used by the microbial community as energy sources. Analysis of the genes encoding the main biochemical pathways showed that carbon and hydrogen cycling are important for microorganisms in this environment and that the serine, RuMP, CBB, WL, and reverse TCA pathways represent the strategies for $CO_2$ fixation. Previous studies reported that all the $CH_4$ emitted from the soil was of abiogenic origin, but the analysis performed in this work revealed a high abundance of a methanogenic archaeon, *Methanocella conradii*. The presence of a hydrogenotrophic methanogen clearly indicates that the methane emissions in the Favara Grande might be a combination of geothermal activity and biological processes taking place in the soil. A follow-up study could focus on detailed analysis of the isotopic composition of the emitted methane. Based on the relative abundance of methanotrophic *Verrucomicrobia*, these are the main $CH_4$-consuming methanotrophs. So far, these methanotrophs have not been detected at this site before using 16S rRNA gene amplicon sequencing techniques. Besides different sequencing methods, this discrepancy could represent an indication of the continuous changes in microbial community that geothermal areas go through over time. The high relative abundance of the methanogenic *Archaea* and the large amount of [NiFe]-evolving hydrogenases indicate that other anaerobic processes may be important in these ecosystems, despite the fact that $O_2$ was measured in the soil. Anaerobic oxidation of $H_2$ while fixing $CO_2$, together with the aerobic oxidation of $CH_4$, might be an important strategy for the establishment of a microbial community. Enrichments using different substrates (methane, hydrogen, etc.) and conditions (aerobic and anaerobic) to isolate key players from the Pantelleria soil and studying their physiology will add to our understanding of this volcanic ecosystem.

## MATERIALS AND METHODS

**Soil and gas sampling.** Gas and soil samples were collected at Favara Grande during a field campaign in June 2017. The locations of the FAV1 and FAV2 sites were described before (6, 22). Soil temperature and pH were measured at 11, 20, 30, and 50 cm (Ebro TFN 520, PT 100 sensors and Hydrion pH 0.0 to 6.0, Brilliant dip sticks). Soil gas samples were taken using a 2-mm-inside-diameter (i.d.) capillary tube at 11, 20, 30, and 50 cm. Gas samples were taken slowly (>60 s to fill a 60-ml syringe) to avoid contaminations with atmospheric gases. Sampling vials (12 ml, Exetainer; Labco Ltd.) were flushed 3 times with 50 ml soil gas and stored at room temperature for further analysis. $CH_4$, $CO_2$, $N_2$, $O_2$, $H_2$, $H_2S$, and He were analyzed by a Perkin Elmer Clarus 500 gas chromatograph (GC) equipped with Carboxen 1000 columns, argon as the carrier gas, and two detectors (hot wire detector [HWD] and flame ionization detector [FID]) (22). Detection limits for gas measurements were 0.1 ppm $CH_4$, 5 ppm He and $H_2$, 50 ppm $O_2$ and $H_2S$, and 100 ppm $N_2$ and $CO_2$. $NH_4^+$ was measured colorimetrically using a modified *o*-phthalaldehyde assay. $NO_2^-$ concentrations were determined using the acid Griess reaction. Before taking the soil samples, the first centimeter of soil was removed. Then, soil samples were taken using a core sampler (diameter, 1.5 cm; depth, 50 cm), and the core was divided into subsections of 5 cm with the exception of the top layer, which contains the top 10 cm of the soil core. Soil samples were stored in sterile 50-ml tubes.

**Nucleic acid extraction.** For metagenomics analysis, DNA was extracted from soil samples of FAV1 and FAV2 divided per depth (1 to 10 cm, 10 to 15 cm, and 15 to 20 cm). To minimize DNA extraction bias, two methods were selected. The Fast DNA Spin kit for soil (MP Biomedicals, Santa Ana, CA), according to the manufacturer's instructions, mechanically lyses the cell material, while in the CTAB DNA extraction, cells are lysed chemically (85). Specifically, cell lysis was obtained by incubating 250 mg of soil from each layer (1 to 10 cm, 10 to 15 cm, and 15 to 20 cm) with 675 $\mu$l of CTAB buffer (100 mM Tris, 100 mM EDTA, 100 mM $Na_2HPO_4$, 1.5 M NaCl, and 1% CTAB), 50 $\mu$l of lysozyme (10 mg/ml; 66,200 U/mg), and 30 $\mu$l of RNase A (10 mg/ml) for 30 min at 37°C. Fifty microliters of proteinase K (10 mg/ml; 20 U/mg) was added to the sample and incubated for 30 min at 37°C. Next, 150 $\mu$l of 10% SDS was added, and the mixture was incubated with for 2 h at 65°C. DNA was extracted by adding 1 volume of phenol-chloroform-isoamyl alcohol (25:24:1) and incubating the sample for 20 min at 65°C. Supernatant was treated with 1 volume of chloroform-isoamyl alcohol (24:1) and centrifuged for 10 min at 20,000 × *g*. Next, 0.6 volume of isopropanol was added to the aqueous phase, and DNA was precipitated by centrifuging the sample for 15 min at 20,000 × *g*. The genomic pellet was washed using ice-cold 70% ethanol and centrifuged for 10 min at 20,000 × *g*. The pellet was air dried and resuspended in 30 $\mu$l of Milli-Q water. DNA was only obtained from FAV2 and not from FAV1. When necessary, DNA samples were purified with DNeasy PowerClean Pro Cleanup kit (Qiagen, Hilden, Germany). The DNA was quantified with a Qubit dsDNA HS assay kit (Thermo Fisher Scientific, Waltham, MA), and it resulted in amounts ranging from 0.2 to 3.8 ng/$\mu$l.

**Metagenomic sequencing.** For library preparation, the Nextera XT kit (Illumina, San Diego, CA) was used according to the manufacturer's instructions. Enzymatic tagmentation was performed starting with 1 ng of DNA, followed by incorporation of the indexed adapters and amplification of the library. After purification of the amplified library using AMPure XP beads (Beckman Coulter, Indianapolis, IN), libraries were checked for quality and size distribution using the Agilent 2100 Bioanalyzer and the high-sensitivity DNA kit. Quantitation of the library was performed by Qubit using the Qubit double-stranded DNA (dsDNA) HS assay kit (Thermo Fisher Scientific, Waltham, MA). The libraries were pooled, denatured, and sequenced with the Illumina MiSeq sequence machine (Illumina). Paired-end sequencing of $2 \times 300$ bp was performed using the MiSeq reagent kit v3 (Illumina) according to the manufacturer's protocol. Illumina MiSeq sequencing resulted in in 17,170,368 (1 to 10 cm), 17,303,428 (10 to 15 cm), and 21,211,358 (15 to 20 cm) reads for DNA extracted using the CTAB method and 23,072,076 (1 to 10 cm), 18,678,982 (10 to 15 cm), and 15,177,544 (15 to 20 cm) reads for the DNA extracted using the Fast DNA Spin kit.

**Phylogenetic analysis.** The read trimming, quality control, and primer removal were performed using BBDuk (86). The trimmed reads were loaded onto the phyloFlash pipeline to retrieve 16S rRNA genes from the metagenome (87). Reads were mapped back to the full-length 16S rRNA genes to calculate relative abundance. For this analysis, the read counts from the different DNA extraction methods were combined. Phylogenetic trees of the 16S RNA gene and *pmoA* sequences were constructed using the maximum likelihood method based on the Tamura-Nei model (88), the tree was bootstrapped (1,000 replicates), and the analysis was performed in MEGA7 (89). The XoxF-type methanol dehydrogenase sequences retrieved from the metagenome were integrated into the data from Keltjens et al. (62), and the phylogenetic tree was built using the maximum likelihood method in MEGA 10.1.7 (90).

**Metagenomic assembly and binning.** The trimmed reads were assembled using MEGAHIT v1.0.3 (91). Reads were aligned to the assembly using Bowtie2 to generate coverage information (92), and files were converted using SAMtools 1.6 (93). The assemblies were binned using different binning algorithms, including BinSanity (94), COCACOLA (95), CONCOCT (96), MaxBin 2.0 (97), and MetaBAT 2 (98). DAS Tool 1.0 was used for consensus binning (99).

**Metabolic reconstruction.** All bins, as well as the unbinned contigs, were loaded into an in-house pipeline to determine the metabolic potential. Custom HMM profiles were generated by downloading protein sequences from the TrEMBL database (https://www.uniprot.org/) corresponding to KEGG numbers (https://www.genome.jp/kegg/) involved in metabolism. The proteins were clustered per K-number using Linclust from the MMSeq2.0 package (100) ("-v 0 –kmer-per-seq 160 –min-seq-id 0.5 –similarity-type 1 –sub-mat blosum80.out – cluster-mode 2 – cov-mode 0 -c 0.7"), and each cluster was aligned using MAFFT v7 ("– quiet –anysymbol") (101) and converted to HMM profiles with hmmbuild (default settings) (60). The hydrogenase profiles were created by downloading all sequences per group from the HydDB website (102) and processed in similar manner. The HMM for ~180 marker genes (see Table S2 in the supplemental material) were manually curated and placed in a separate database. Prodigal ("-m -c -g 11 -p single -f sco -q") was used for gene calling (103), after which, these custom HMM profiles were used to find the 504 metabolic key genes in the metagenome. The remaining HMM profiles were used to annotate the other metabolic genes and default Prokka settings were used to further annotate the metagenome-assembled genomes (MAGs) (104).

**Data availability.** DNA sequences (raw sequence reads and MAGs) have been deposited in the NCBI BioProject database with project number PRJEB36447. NCBI GenBank accession numbers for individual MAGs are listed in Table S3.

## SUPPLEMENTAL MATERIAL

Supplemental material is available online only.

**FIG S1**, PDF file, 0.1 MB.

**FIG S2**, PDF file, 0.5 MB.

**TABLE S1**, PDF file, 0.1 MB.

**TABLE S2**, PDF file, 0.1 MB.

**TABLE S3**, PDF file, 0.1 MB.

## ACKNOWLEDGMENTS

C.H., N.P., H.J.M.O.D.C., and T.B. were supported by the European Research Council (ERC advanced grant project VOLCANO 669371), L.P. was supported by the Netherlands Earth System Science Centre (NESSC 024002001), and M.S.M.J. was supported by the European Research Council (ERC advanced grant project Eco_MoM 339880) and The Soehngen Institute of Anaerobic Microbiology (SIAM 024002002).

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
