## [Reviewer comments · mSystems]

Geothermal gases shape the microbial community of the volcanic soil of Pantelleria, Italy

Nunzia Picone, Carmen Hogendoorn, Geert Cremers, Lianna Poghosyan, Arjan Pol, Theo van Alen, Antonina Gagliano, Walter D'Alessandro, Paola Quatrini, Mike Jetten, Huub Op den Camp, and Tom Berben

Corresponding Author(s): Huub Op den Camp, Radboud University Nijmegen

Review Timeline:

Submission Date:	June 8, 2020
Editorial Decision:	July 6, 2020
Revision Received:	October 12, 2020
Accepted:	October 14, 2020

Editor: Karen Lloyd

Reviewer(s): Disclosure of reviewer identity is with reference to reviewer comments included in decision letter(s). The following individuals involved in review of your submission have agreed to reveal their identity: Gerhard L Jessen (Reviewer #1)

Transaction Report:

DOI: <https://doi.org/10.1128/mSystems.00517-20>

July 6, 2020

Prof. Huub J.M. Op den Camp
Radboud University Nijmegen
Department of Microbiology
Faculty of Science
Heyendaalseweg 135
Nijmegen NL-6525 AJ
Netherlands

Re: mSystems00517-20 (Microbial community and metabolic potential of the volcanic soil of Pantelleria, Italy)

Dear Prof. Huub J.M. Op den Camp:

The modifications suggested by the reviewers all need to be addressed before this manuscript can be accepted. A few are substantial, and may require significant manuscript alterations. But I believe this can be accomplished without generating new data or a complete re-write of the manuscript. I highlight a few of the key modifications below:

The introduction should highlight the ways in which the current work is different than previous work done at this site, and which research questions are addressed here that have not been previously answered.

The quantifications need to be relative to total biomass, either with cell counts or total extracted DNA. These need to take into account the fact that some pathways have more or longer genes than others. See reviewer #2's comments for details.

Although the methods say that binning was performed, the bulk of the results presented are from metagenome assemblies. Either introduce and describe the bins in the results section, and discuss them, or take the description of binning out of the methods section, since no results are presented from it. In general, it needs to be clearer when data were derived from 16S amplicons, or from either genes or bins from the metagenomes. For instance: Fig. 1 should say whether this is amplicon or 16S derived from metagenomes. And, when you begin talking about metagenomic results (line 215), it's not clear whether this is from assembled metagenomic data or from bins. Again, on line 219, it is unclear whether the 7-12% refers to amplicons or metagenomic bin read recruitment.

Below you will find the comments of the reviewers.

To submit your modified manuscript, log onto the eJP submission site at <https://msystems.msubmit.net/cgi-bin/main.plex>. If you cannot remember your password, click the "Can't remember your password?" link and follow the instructions on the screen. Go to Author Tasks and click the appropriate manuscript title to begin the resubmission process. The information that you entered when you first submitted the paper will be displayed. Please update the information as necessary. Provide (1) point-by-point responses to the issues raised by the reviewers as file type "Response to Reviewers," not in your cover letter, and (2) a PDF file that

indicates the changes from the original submission (by highlighting or underlining the changes) as file type "Marked Up Manuscript - For Review Only."

Due to the SARS-CoV-2 pandemic, our typical 60 day deadline for revisions will not be applied. I hope that you will be able to submit a revised manuscript soon, but want to reassure you that the journal will be flexible in terms of timing, particularly if experimental revisions are needed. When you are ready to resubmit, please know that our staff and Editors are working remotely and handling submissions without delay. If you do not wish to modify the manuscript and prefer to submit it to another journal, please notify me of your decision immediately so that the manuscript may be formally withdrawn from consideration by mSystems.

To avoid unnecessary delay in publication should your modified manuscript be accepted, it is important that all elements you upload meet the technical requirements for production. I strongly recommend that you check your digital images using the Rapid Inspector tool at <http://rapidinspector.cadmus.com/RapidInspector/zmw/>.

Sincerely,

Karen Lloyd

Editor, mSystems

Journals Department
Reviewer comments:

Reviewer #1 (Comments for the Author):

The manuscript submitted by Picone and colleagues assessed the microbial community and metabolic potential of the volcanic soil of Pantelleria, Italy. Picone and colleagues aimed to link microbially available geogases to the metabolic potential of the microbial community. Combining metagenomics and geochemical approach, the authors found that the methane emissions in the

Favara Grande might result from geothermal activity and biological processes.

Below I would like to address a set of general and specific comments and questions that may be considered by the authors to further improve this manuscript.

Although I do like the title, I think it's too descriptive and would gain from a more active voice.

Introduction is not well structured. I would suggest follow the structure written in L114-120.

Results and discussion are overall logic, but the structure is not clear beyond description or according to the "importance section" of the research. I'm missing thoughts and literature reviews in the context of the objectives and main findings. This is more evident in case of methanogenesis.

Statistical analyses are absent, although I understand the limitations due to sample size.

The English language is correct, but I would recommend a proof reading.

Tables and figures are of good quality, but phylogenetic trees are not fully discussed and tables 2 and 3 can be merged. Indeed, Sup Fig 1 is more relevant for the story as main figure, than the trees.

Please specify when a sample was not measured and the detection limit in Table 1. Also, sampling scheme needs to be consistent, habitat and community are inconsistent as presented (i.e. discrete horizons in Table 1 and sections for the rest). Methods section needs to clarify the sampling design and sample collection.

Specific comments:

L27: Add the aim of the study

L28: if depth is important include it

L46: the unique community might be shaped by the hostile conditions

L57: Here and later, clarify the criteria to list the gases 3

L67: Hard to understand the system from those names, consider a short description of the sampling design

L75-77: Please re-write for clarity

L128: This was the result of one sample? if so, any idea about the variability of the system considering the marked differences from those two seeps only meters apart?

L148: (and the geochemical description). Is confusing the criteria used to describe the geochemistry

L153: how the authors separate one mechanism of the other (i.e. biological activity vs. physical processes)?

L157: below detection limit (i.e. 50 ppm). Also, please discuss He dynamics

L160: As before, I wonder how representative Table 1 is

L166: Mentioned before, please specify the actual horizon. Also, how samples were merged and when results come from kit vs. CTAB extraction. Discuss.

Fig. 1: "Domain" not Kingdom. Please specify the exact horizon from where the sample was retrieved. Was all the sediment homogenized per section? Like this it's hard to link the community results to the geochemistry. Also, those were discrete samples, I would suggest to use the actual horizon (as in table 1).

L177: what cutoff? > XXX%? present everywhere? please explain

L191: Use the right chemical zonation (see Canfield & Thamdrup, Geobiology (2009), 7, 385-392)

L204: A whole section is dedicated to nitrogen cycling, even if the data is not shown I would suggest to provide a range.

L205: provide a diversity index

Community analysis section: I'm not getting the effort to link function with taxonomy having its actual metabolic potential from the metagenomes

L222: this might be an issue in this study too, please address that

Figure 2: add "not detected" or something to the legend to explain the white squares. Also, this figure is not fully discussed, either do it or move it to Supplementary section (same for fig. 4). The

colored triangles depict coverage and extraction method

L251: In the core here studied. FAV1 has high H₂S concentrations.

Fig 3. This figure could be turned into a schematic model linking the geochemical data, overall providing a better visualization of the results

L266. Methane section: Methanogenesis is not discussed at all and play a major role according to the story and geochemistry.

L410: oxidation rather than cycling

L423-426: is not well addressed in the text

L438: Describe briefly sampling design and collection

L443: Provide detection limits

L451: since different extractions methods were used would be worth to mention any difference.

Please specify what sample was extracted with what method

L455: from which horizon? were all merged? how was the sediment sampled?

L493: different trees and software versions, please clarify

Reviewer #2 (Comments for the Author):

Thank you for the opportunity to review this manuscript: I applaud and appreciate the substantial effort that has gone into acquiring these data and conducting the associated analysis. My comments are submitted with the sole intention of improving the manuscript and its impact.

Recap & Summary:

In their manuscript, "Microbial community and metabolic potential of the volcanic soil of Pantelleria, Italy," Picone et al. offer a metagenomic analysis of the microbial community in geothermal sediment. With the inclusion of physicochemical data, the researchers link the presence of gaseous species with 16S rRNA and metabolic marker genes to propose a model of elemental cycles. In two sediment cores, they report variable degree of mixing between H₂-, CH₄-, and CO₂-rich subsurface gases with O₂- and N₂-rich atmospheric gases. Metagenomic analysis was only performed for one of the cores. Community analysis, elemental cycling, and carbon fixation processes were inferred from the presence and relative abundance of function genes. Phylogenetic analyses of methane monooxygenase, methanol dehydrogenase, and hydrogenase demonstrated a diversity of organisms likely conducting metabolic processes of interest. Overall, the manuscript offers a window into the physicochemical conditions in upper horizons of volcanic soils, and a cursory glimpse into the identity and function of the microbial community. This study nicely conveys the complexity and potential dynamism of the community at this intriguing boundary zone, but the interpretive power of the dataset as presented is limited due to a lack of substantive engagement with previous work, imprecise (and potentially misleading) quantification efforts, and the seemingly unnecessary aggregation of data across spatially-resolved soil horizons.

Major Comments:

1. The work presented here represents an incremental step beyond previously published work from many of the same authors (D'Alessandro et al., 2009, Gagliano et al., 2014, and Gagliano et al., 2016). These previous works contain more detailed assessments of the site's volatiles, microbial diversity, metabolic rates, and community composition at both FAV1 and FAV2. I could envision two potential pathways to bolster this manuscript's contribution and move beyond previously published data. One would be to build a more detailed and coherent metabolic model, linking lineage-specific functional genes along complete metabolic pathways (not only marker genes) to propose a more

complete biogeochemical network and highlight putative inter-organism relationships. Another would be to more directly compare this work with the 2009, 2014, and 2016 papers and engage with the discrepancies to develop a more nuanced understanding of the site and its temporal variations. For example, why do both FAV1 and FAV2 show evidence of atmospheric mixing in their upper horizons in the 2016 paper, but only FAV2 does in this manuscript? How does the 2014 pmoA gene library compare with the one generated here? If the authors were to more directly engage the temporal component of the data collected over the last decade, a deeper understanding of how volcanic emissions and microbial processing change with time could be developed.

2. The quantification efforts - in both the 16S rRNA genes (Fig. 1) and the carbon and hydrogen cycling (Fig. 3) - present some problems. The 16S data would be more representative of the community structure and influence if it were scaled by the overall biomass for each horizon. As currently presented, the figure (and all aspects of the text) suggest that there was equal abundance of microbial cells at all horizons. Was this in fact the case? While cell counts would probably be the best way to derive appropriate scaling factors, such data may not be available; the quantified extracted DNA from the Qubit would be a decent back-up option. By scaling the relative abundance by the actual abundance, the authors can provide a more realistic sense of how communities at different horizons compare. For example, aspects of the "community analysis" section could change substantially; e.g., statements such as "the amount of Methylococcaceae remained more or less constant."

Of greater importance, perhaps, is the quantification of functional genes and pathways; I did not find statements such as "13.1% of the retrieved genes encode for proteins involved in the CH₄ oxidation pathway" particularly useful. This is because the relevant denominator isn't well-constrained. For example, if one pathway involves 20 genes, and another involves 2, one might expect many more of the recovered genes to map to the first pathway even if the second pathway is recovered in its entirety. Gene length should also be taken into account in these assessments. In addition, the use of genes for multiple pathways (see comment on mdh below) could result in over-counting of pathway representation.

3. Lines 479-482 indicate that separate metagenomes were acquired for each horizon: 0-10, 10-15, and 15-20cm. If this is the case, then why were the metagenomes - and all of the associated functional genes and metabolic pathways - not analyzed separately, by horizon? Given the vertically resolved geochemical and 16S analyses, there is much to be gained by disaggregating the metagenomic data. Does the relative abundance of methanotrophy, or various carbon fixation pathways, change with depth? The clear vertical redox zonation shown in Table 1 offers a compelling lens through which to view putative metabolic pathways.

4. The reliance on marker genes as representatives of entire pathways seems unnecessarily limiting. In my experience, marker genes are typically used with specific primers, when metagenomes are not available. In the case of this manuscript, a metagenome should in theory allow for reconstruction of full pathways. In the discussion on methanotrophy and methylotrophy, for example, why did the analysis not include formaldehyde dehydrogenase and formate dehydrogenase? A more thorough analysis of full pathways could expose metabolic bottlenecks, potential metabolic crosstalk, and patterns in gene copy number. When normalized by genome coverage and gene length, it would also help make any quantitative treatments based on the data more credible.

Additional Comments:

Lines 28-31: The references to "top layer," "increases with depth," and "deeper layers" are not particularly useful without reporting what those depths are. Perhaps the "various depths" mention in line 27 could include a parenthetical note on which depths were sampled for community structure.

Lines 37-39: I'm not sure the percentage values for carbon fixation pathways are useful or necessary in the abstract; what are these percentages of? It might be best to leave the percentages for the main text where the method for determining percentages can be better explained.

Introduction: Additional background on the field site would help place this work in a more holistic geobiological context. For example, what type of volcano is Pantelleria, how long has it been active, and how do its emissions (content and quantity) compare with other volcanoes? Why was this site chosen for this work? How were FAV1 and FAV2 selected - why were more sites not sampled? A figure showing the field site would also be very helpful. This figure could include a map, images from the field, and labels of where precisely the two soil cores were collected.

Line 66: Please provide a reference for this sentence.

Line 90: This reference pertains to research conducted in anoxic submarine sediments, which are not so analogous to the subaerial systems being discussed in this manuscript and implied in this paragraph. At marine mud volcanoes, methane is almost entirely consumed either in the sediment or water column, and thus are less likely than subaerial volcanic systems to "contribute to the world's CH₄ emissions."

Lines 102-107: This list of carbon fixation pathways does not, in my opinion, add much to the introduction and could be removed. In its place, perhaps the authors could describe and explain the carbon fixation pathways typically found at geothermal environments. Do such environments shape carbon fixation approaches in ways that are substantively different from "normal" soils or less selective environments?

Lines 114-116: Previous work at this site has largely "determine[d] the important gaseous electron donors supporting the chemolitho(auto)trophic microbial community," which is the stated goal of this manuscript. Perhaps the authors could better explain how this study goes beyond the earlier work. The focus on hydrogen could be helpful, and lines 118-120 are good, but it would be nice to see a more clear distinction from previous work in the stated goals. (If things are substantially restructured based on major comment #1 above, the stated objectives could also include a longer-term analysis of how the microbial community changes at this site.)

Lines 128-130: This assertion is not well supported, and could be interpreted as contradicting the later suggestion - and foundational premise of this study - that the microbial community influences the gas content. Relevant references would be appreciated here, and a bit more clarity on the idea being put forth. Are the hydrothermal gases flowing upward from different reservoirs? Do they react along the way in a manner that changes their composition? And if so, then aren't the differences due more to what is encountered along the way rather than simply a different path?

Lines 131-136: References and a little more detail would be helpful here. What kinds of gas-water-rock interactions are relevant here? How exactly do depletions and enrichments happen? How is permeability lowered - are there precipitation reactions that occur?

Lines 141-142: The production of sulfuric acid via sulfide oxidation as suggested in ref. 12 is intriguing, but I wonder if there is any evidence of this. It doesn't seem that any known sulfide oxidizers or epsilonproteobacteria were found at FAV1 previously...

Lines 142-143: It is often challenging to link concentration of a metabolite to putative metabolic activity using said metabolite. For example, higher sulfide concentrations could indicate that sulfide production is simply faster than sulfide consumption, but both values could be very high.

Lines 149-153: The dueling interpretations of O₂ and N₂ increases and H₂ and CH₄ decreases at the surface lack consistency. If atmospheric air is permeating the soil to increase O₂ and N₂, why wouldn't it simultaneously be diluting H₂ and CH₄? The biological activity explanation is certainly a possibility, but at this point in the discussion, I'm not sure we can rule out the abiotic scenario.

Lines 154-156: The preferential decrease of H₂ is an interesting finding; in addition to the biological consumption possibility, do the authors think its smaller molecular size and higher mobility through porous media could be a relevant factor?

Lines 168-169: Some additional information on the use of two extraction methods would be useful, either here or in the methods section. What are the specific known biases of these two methods? Do the two methods complement each other in that sense, making up for the other's blind spots? If you were to disaggregate the 16S data, did the effort to avoid biases work? (This is addressed with the methanotrophs in lines 236-239 and Fig. 2, but extending such an analysis beyond those lineages would be very helpful.)

Lines 172-173: 62 16S rRNA genes seems surprisingly low. I realize this is because they were derived from the metagenomic data and they are full length, which helps provide more taxonomic resolution, but might the authors comment on this number, especially in comparison to the 2016 paper, where FAV2 produced 147 OTUs? Is there a reason an additional high-throughput 16S-specific sequencing run was not performed?

Lines 181-183: I agree this is surprising, given the earlier measurements of methane's $\delta^{13}\text{C}$ values of -17.5 and -17.2% , which is far from the traditionally accepted biogenic range of values. How do the authors reconcile their detection of methanogens with the isotopic measurements from D'Alessandro et al.? What kind of mixing ratio might be required to allow for methanogenesis but retain these heavy carbon values?

Lines 189-199: This section on attempts to reconcile methanogenic activity with aerobic soils is great - it succinctly explains the challenges, cites important recent studies on the subject, and proposes a possible explanation.

Lines 202-204: Is there a reason these data on ammonium concentrations are not shown? I think they would be very helpful in allowing the reader to compare the site to other locations and evaluate the biogeochemical impact the ammonia oxidizers could be having.

Line 205: This assertion seems reasonable based on figure 1, but it might be more convincing to provide alpha diversity statistics of each horizon and the archaeal and bacterial communities associated with each. This information would also help unpack just how much more diverse the bacterial community was.

Lines 214-215: This sentence is somewhat misleading: "this study" refers to the Gagliano 2016

study (right?), but could be interpreted as the Picone et al. manuscript itself.

Lines 217-222: This section is interesting, and the new detection of a more substantial verrucomicrobial component represents an important result in comparison to past analyses. Does the PowerSoil kit (which I believe was used in Gagliano et al., 2016?) specifically miss Verrucomicrobia?

Lines 232-234: A brief account of the gammaproteobacterial aerobic methanotrophs' acid tolerance would be welcome here, in order to bolster the point that pH may be a selective factor. Islam et al., *Frontiers in Microbiology*, 2016 could be a good place to start.

Lines 274-277: The specific analysis of *pmoA* vs. *amoA* is very helpful, and the lack of *amoA* sequences is unexpected given the presence of the Thaumarchaea. How do the authors explain this discrepancy? Could the relative coverage of the thaumarchaeal genomes help? For example, if the genome is only 5% closed, one might not expect to find the *amoA* gene. Comparing coverage of the methanotrophs with the Thaumarchaea - as well as the gene copy number and lengths of their respective functional genes - could be interesting.

Lines 281-284 and throughout: For all of the functional genes, it would be nice to know how many of them mapped onto contigs that also had a 16S rRNA gene. In other words, how many of these taxonomic inferences of functional genes are relatively certain, and how many are circumstantial?

Figure 4: Perhaps, as with figure 2, the taxonomic assignments could be placed alongside the vertical colored bars on the right; this saves the reader a step, and eliminates the need for a legend.

MDH discussion: Since MDHs can be used by non-methanotrophs such as methanol oxidizers, how do we know the 46 recovered *mdh* genes are only being used for methanotrophy?

Table 3: What could the offset between the number of genes and the relative abundance of genes be indicating? Do any of these hydrogenases map back onto a 16S rRNA gene that could help establish a link between gene : relative abundance ratio and relative abundance of the associated organism?

Lines 355-357: Here, as well as with the discussion of methanogens in aerated soils and throughout the manuscript, it might be helpful to consider that the presence and recovery of genes does not mean that the associated enzyme or pathway are active. Could it be possible that these genes are left over from a time when conditions were anoxic? Or that the associated organisms were transported from anoxic zones? Is the gas emission strong enough to transport organisms from depth into the upper reaches of the soil column?

Lines 358-362: Listing other sites of hydrogenase recoveries is not particularly informative without an interpretation. For example, what is it about "forest soil and permafrost" - the microbial communities, the environmental parameters - that could explain why organisms with NiFe-hydrogenases are found both there and in the Favara Grande samples?

Lines 364-392: This section strikes me as a list of carbon fixation pathways that could benefit from additional discussion and interpretation. For example, lines 372-375 is so high-level as to offer very little insight. I'm not sure exactly what the best way to enhance this section would be, but two options could include the following. 1) Getting more granular to see how specific lineages process carbon and could interact directly or indirectly with other organisms. 2) Getting even higher-level to

see how this distribution of carbon fixation pathways within a given microbial ecosystem compares with other types of systems. And what would the diversity of carbon fixation approaches tell you about the system and its redox zonation, energy availability, available niches?

Lines 395-400: I'm not sure I understand the reasoning here. If there "is probably not much of a niche for ammonia oxidizers," then how were so many recovered via 16S rRNA analysis? Perhaps a different interpretive track to pursue is the possibility of a low standing stock of ammonium - that it may be produced or introduced, but that it's consumed very quickly.

Lines 403-404: Why was nitrogenase activity "most likely involved in N₂-fixation for biomass production"?

Discussion / Conclusion: What are some concrete next steps this research could take to address some of the (many) remaining questions raised in this manuscript? For example, the source of the methane, the ammonium paradox, the DNA recoverability issues, etc.

Line 417: Perhaps "emissions" should be replaced by "concentrations," as no flux measurements are reported in this manuscript.

Lines 427-428: Presumably you mean they "have not been detected" at this site before?

Lines 438-441: What was the reasoning behind the selection of sampling depths in this study? Why 11 cm and not 10, and why 30 and 50 but not 40? Why not deeper, or more compressed (e.g., 5 cm) horizons?

Lines 439-440: A little more detail on the use of pH strips might be helpful. What is the margin for error with these? If it's what I'm envisioning, you match the color to the closest 0.5 pH unit, which suggests that if the closest color is 4.0, then the actual pH is between 3.75-4.25? Were the user's visual approximations calibrated in the lab with a pH probe? Doing so would help establish the reported measurements as more trustworthy.

Lines 441-443: I find a couple of items in this description slightly confusing. First, when collecting 60 mL of gas, can you calculate the expected "sphere of influence" from which this gas was drawn? If we presume equal sucking power in all directions, this can be calculated by measuring the porosity at each horizon. This would give a range of depths that would more accurately represent the situation; the "30 cm" horizon might in fact be, 28-32 cm or something. Second, what kind of gas were the sampling vials flushed with? If it was gas from the horizon of interest, it seems, as written, that 150 mL of gas were used for this...and yet only 60 mL were collected? Please clarify both of these points.

Lines 445-446: Why was the top cm of soil removed from the analysis, and how do the authors imagine this changed their results? I would expect a distinct, photosynthetically driven community at the air-soil interface - omitting this horizon from the analysis likely changed the diversity results substantially.

Line 452: To clarify, were gas and DNA sampled collected from the same core? If so, how was contamination from inserting the capillary tube at different depths ruled out? If not, this should be mentioned, and the potential repercussions should be addressed. For example, given the heterogeneity found between FAV1 and FAV2 ("only a few meters apart") how would taking parallel cores change the community structure and metabolic interpretations? (Incidentally,

Gagliano et al., 2016 described FAV1 and FAV2 as being "about 10 m apart;" please explain this inconsistency.)

Line 466 and throughout: I found it slightly distracting that the horizons were labeled inconsistently as "top layer," "10-15 cm," and "15-20 cm." It might be easiest to indicate what the top layer depth range was. Earlier I believe it was specified as 0-10 (line 167)...which, in reference to the comment above, suggests the geochem and molecular work was done on separate cores?

Lines 465-468: The failure to recover DNA from FAV1 is perplexing, and additional troubleshooting efforts or explanations would be welcome. Previous work was able to recover sequenceable DNA from this site; why was this effort less successful, despite using two extraction methods and, it appears, having more material to work with (10 cm horizons vs. 0-3 in Gagliano et al., 2016)? To what do the authors attribute the poor extractions? Soil chemistry, lack of biomass, etc.? Were cell counts conducted to double-check the relative amounts of biomass? I would also be curious to know what the DNA quantification results showed, and believe this is important for overall interpretation of the results (see major comment 2 above).

Lines 497-499: I am admittedly not a metagenomics expert, but this seems like a lot of binning algorithms! Why were all of them necessary? Did they all agree with each other? How was data from one algorithm used to seed the next algorithm? Further explanation of why so many approaches were used - and why each one was necessary - would be appreciated.

Lines 501-504: More detail on the metabolic reconstruction approach is needed to allow the reader to fully understand and evaluate what happened. What is involved in the "in-house" pipeline? How were the customized HMM profiles developed and validated? What settings were used for Prodigal and PROKKA? (In the previous section, what were the specific settings for all of the assembly and binning algorithms?)

Rebuttal mSystems00517-20 R0

Comments of the Editor

The introduction should highlight the ways in which the current work is different than previous work done at this site, and which research questions are addressed here that have not been previously answered.

The previous work was amplicon-based while our current work is full metagenome sequence-based. Amplicon sequencing only involves part of the 16S rRNA genes. On the original L108-113 we introduced the advantages of using metagenomics rather than 16S amplicon. To emphasize this even more, we have put the sentence starting with "It remains difficult to link 16S rRNA gene amplicon..." in the active voice. We hope this clarifies that we used metagenomics and not 16S amplicon sequencing.

The quantifications need to be relative to total biomass, either with cell counts or total extracted DNA. These need to take into account the fact that some pathways have more or longer genes than others. See reviewer #2's comments for details.

Soil cell counts are impossible in these soils. Metagenomic studies do not involve cell counts but more refers to the amount of DNA extracted. However, still comparison between different soil types may be difficult. We have added the DNA amounts extracted to the manuscript.

Although the methods say that binning was performed, the bulk of the results presented are from metagenome assemblies. Either introduce and describe the bins in the results section, and discuss them, or take the description of binning out of the methods section, since no results are presented from it. In general, it needs to be clearer when data were derived from 16S amplicons, or from either genes or bins from the metagenomes. For instance: Fig. 1 should say whether this is amplicon or 16S derived from metagenomes. And, when you begin talking about metagenomic results (line 215), it's not clear whether this is from assembled metagenomic data or from bins. Again, on line 219, it is unclear whether the 7-12% refers to amplicons or metagenomic bin read recruitment.

There is no amplicon sequencing in our story (see above), the 16S rRNA gene data used for Figure 1 are extracted from the metagenome sequencing data. We changed the wording of this part. After the primary assembly of the metagenome reads (without binning) all contigs were searched for genes involved in crucial pathways (gene-driven). In addition the binning supplies information on the microorganisms present an read coverage.

Reviewer #1 (Comments for the Author):

The manuscript submitted by Picone and colleagues assessed the microbial community and metabolic potential of the volcanic soil of Pantelleria, Italy. Picone and colleagues aimed to link microbially available geogases to the metabolic potential of the microbial community. Combining metagenomics and geochemical approach, the authors found that the methane emissions in the Favara Grande might result from geothermal activity and biological processes.

Although I do like the title, I think It's too descriptive and would gain from a more active voice.

We have change the title into : "Geothermal gases shape the microbial community of the volcanic soil of Pantelleria, Italy"

Introduction is not well structured. I would suggest follow the structure written in L114-120.

L114-120 only describes aim and methods. In our introduction we start broadly describing geothermal environments, then we move to the description of our sampling site (Pantelleria) followed by a description of previous work and metabolisms encountered in volcanic environments. We then talk about the limitation of the previous 16S rRNA amplicon sequencing analysis and why implementing metagenomics would improve the description of the microbial community. Finally we conclude with the aim of the study and with a small summary of the experiments performed.

Results and discussion are overall logic, but the structure is not clear beyond description or according to the "importance section" of the research. I'm missing thoughts and literature reviews in the context of the objectives and main findings. This is more evident in case of methanogenesis.

The observation of methanogenesis in comparable ecosystems was not made before. The physiological information on the only cultured representative *M. conradii* HZ254 is used to discuss the potential for methanogenesis in volcanic ecosystems.

Statistical analyses are absent, although I understand the limitations due to sample size.

We have added statistical information to Table 1. For Figure 1 it is not possible to calculate SD since distribution of the phyla included data from both extraction methods. However, we have now added a Supplementary Table S1 reporting the individual values for each DNA extraction method. In addition Figure 3 is not quantitative.

The English language is correct, but I would recommend a proof reading.

We carefully checked the manuscript for use of the English language.

Tables and figures are of good quality, but phylogenetic trees are not fully discussed and tables 2 and 3 can be merged.

We have added some more detailed discussion on the phylogenetic trees. The figure was changed and top layer substituted with 1-10 cm. The two tables deal with completely different enzymes so we would prefer to keep them as separate tables.

Indeed, Sup Fig 1 is more relevant for the story as main figure, than the trees.

Supplementary Fig. S1 is very detailed, to our opinion may be much too detailed for the main text. However, we decided to follow the suggestion of the reviewer and moved this figure to the main text (now Fig. 4). Original Figure 4 renumbered to 5.

Please specify when a sample was not measured and the detection limit in Table 1. Also, sampling scheme needs to be consistent, habitat and community are inconsistent as presented (i.e. discrete horizons in Table 1 and sections for the rest). Methods section needs to clarify the sampling design and sample collection.

The profiles for Table 1 were possible on a higher resolution compared to the metagenomic analysis, Detection limits are added in the methods section. Name of the sampled layers were consistently changed throughout the manuscript.

Specific comments:

L27: Add the aim of the study

We have added this.

L28: if depth is important include it

Included

L46: the unique community might be shaped by the hostile conditions

Sentence modified as requested.

L57: Here and later, clarify the criteria to list the gases 3

Concentrations of the gases are variable. We have included this were needed.

L67: Hard to understand the system from those names, consider a short description of the sampling design.

We have rephrased this part. For detailed description two references are cited.

L75-77: Please re-write for clarity

Re-written as requested to improve clarity.

L128: This was the result of one sample? if so, any idea about the variability of the system considering the marked differences from those two seeps only meters apart?

We have added a more detailed description.

L148: (and the geochemical description). Is confusing the criteria used to describe the geochemistry

The sentence was re-formulated.

L153: how the authors separate one mechanism of the other (i.e. biological activity vs. physical processes)?

We changed the sentence: "This counter gradient could enable biological activity."

L157: below detection limit (i.e. 50 ppm). Also, please discuss He dynamics

We have extended this part of the manuscript.

L160: As before, I wonder how representative Table 1 is

We included analytical errors in the legend.. Our focus is on metagenomics and the Table is representing a snapshot for the time of soil sampling. More extensive analysis is out of scope of the current manuscript.

L166: Mentioned before, please specify the actual horizon. Also, how samples were merged and when results come from PS kit vs. CTAB extraction. Discuss.

Horizons are specified multiple times in methods and main text. In this specific case they are listed in L167. The confusion may come from gas sampling (deepest at 50 cm) in comparison to DNA extraction (deepest at 20 cm). "Top layer" has been changed into "1-10 cm". Differences of the extraction methods are shown in the 16S rRNA gene analysis (tree). To avoid extraction bias we combined the reads of the DNA from both methods after sequencing.

Fig. 1: "Domain" not Kingdom. Please specify the exact horizon from where the sample was retrieved. Was all the sediment homogenized per section? Like this it's hard to link the community results to the geochemistry. Also, those were discrete samples, I would suggest to use the actual horizon (as in table 1).

We corrected Fig. 1. Yes, sediments were homogenized per section otherwise we could not obtain enough DNA to perform metagenomics.

L177: what cutoff? > XXX%? present everywhere? please explain

The legend of Fig. 1 has been modified to explain this.

L191: Use the right chemical zonation (see Canfield & Thamdrup, Geobiology (2009), 7, 385-392)

We changed "(semi)aerobic environments" into "environments with low oxygen levels".

L204: A whole section is dedicated to nitrogen cycling, even if the data is not shown I would suggest to provide a range.

We included concentrations of nitrogen compounds measured.

L205: provide a diversity index Community analysis section: I'm not getting the effort to link function with taxonomy having its actual metabolic potential from the metagenomes

We made a small chart of the Simpson index (alpha diversity) of the three horizons split up between bacteria and archaea based on the phyloflash data and included this as Supplementary Fig. S1.

L222: this might be an issue in this study too, please address that Figure 2: add "not detected" or something to the legend to explain the white squares. Also, this figure is not fully discussed, either do it or move it to Supplementary section (same for fig. 4). The colored triangles depict coverage and extraction method

The legend of Fig. 2 was modified. more details were added to the text.

L251: In the core here studied. FAV1 has high H₂S concentrations.

Sentence was reformulated.

Fig 3. This figure could be turned into a schematic model linking the geochemical data, overall providing a better visualization of the results

The geochemical data as already indicated are a snapshot for the soil sampling. This figure is a schematic model and we did our best to make it as comprehensive and understandable as possible. We would need flux measurement to make real sense of a coupling with geochemical data.

L266. Methane section: Methanogenesis is not discussed at all and play a major role according to the story and geochemistry.

Methanogens show a high relative abundance. Difficult to say how active they are at the moment of sampling. In times of low O₂ they might become active. Conditions might change in these ecosystems. (see also above)

L410: oxidation rather than cycling

Modified accordingly.

L423-426: is not well addressed in the text

It is difficult to discuss since the observation of methanogenesis was not made before.

L438: Describe briefly sampling design and collection

We did our best to describe sampling design and collection. The location FAV1 and FAV2 are extensively described in the reference cited.

L443: Provide detection limits

Detection limits are provided.

L451: since different extractions methods were used would be worth to mention any difference.

Please specify what sample was extracted with what method

These differences are already illustrated in the Figure 16S tree (Fig. 2). More description has been added to the legend, All samples that were analyzed were extracted with the two different methods as indicated in the Materials and Methods section.

L455: from which horizon? were all merged? how was the sediment sampled?

Samples were not physically merged. DNA was extracted and sequenced separately. Reads were merged together. More details added in the text. See also comments Reviewer 2.

L493: different trees and software versions, please clarify

The basic software (algorithms) is not different. Two versions of the phylogenetic analysis software MEGA were used and cited accordingly.

Reviewer #2 (Comments for the Author):

Thank you for the opportunity to review this manuscript: I applaud and appreciate the substantial effort that has gone into acquiring these data and conducting the associated analysis. My comments are submitted with the sole intention of improving the manuscript and its impact.

Recap & Summary:

In their manuscript, "Microbial community and metabolic potential of the volcanic soil of Pantelleria, Italy," Picone et al. offer a metagenomic analysis of the microbial community in geothermal sediment. With the inclusion of physicochemical data, the researchers link the presence of gaseous species with 16S rRNA and metabolic marker genes to propose a model of elemental cycles. In two sediment cores, they report variable degree of mixing between H₂-, CH₄-, and CO₂-rich subsurface gases with O₂- and N₂-rich atmospheric gases. Metagenomic analysis was only performed for one of the cores. Community analysis, elemental cycling, and carbon fixation processes were inferred from the presence and relative abundance of function genes. Phylogenetic analyses of methane monooxygenase, methanol dehydrogenase, and hydrogenase demonstrated a diversity of organisms likely conducting metabolic processes of interest. Overall, the manuscript offers a window into the physicochemical conditions in upper horizons of volcanic soils, and a cursory glimpse into the identity and function of the microbial community. This study nicely conveys the complexity and potential dynamism of the community at this intriguing boundary zone, but the interpretive power of the dataset as presented is limited due to a lack of substantive engagement with previous work, imprecise (and potentially misleading) quantification efforts, and the seemingly unnecessary aggregation of data across spatially-resolved soil horizons.

Thank you for the efforts to improve our manuscript. This reviewer is as he/she states "I am admittedly not a metagenomics expert". It is very important to understand the big difference between metagenome approaches and amplicon sequencing. In amplicon sequencing only a small part of the 16S rRNA genes are amplified and sequenced. The metagenome approach includes random sequencing of all DNA extracted without primer-driven PCR amplification.

Major Comments:

1. The work presented here represents an incremental step beyond previously published work from many of the same authors (D'Alessandro et al., 2009, Gagliano et al., 2014, and Gagliano et al., 2016). These previous works contain more detailed assessments of the site's volatiles, microbial diversity, metabolic rates, and community composition at both FAV1 and FAV2. I could envision two potential pathways to bolster this manuscript's contribution and move beyond previously published data. One would be to build a more detailed and coherent metabolic model, linking lineage-specific functional genes along complete metabolic pathways (not only marker genes) to propose a more complete biogeochemical network and highlight putative inter-organism relationships. Another would be to more directly compare this work with the 2009, 2014, and 2016 papers and engage with the discrepancies to develop a more nuanced understanding of the site and its temporal variations. For example, why do both FAV1 and FAV2 show evidence of atmospheric mixing in their upper horizons in the 2016 paper, but only FAV2 does in this manuscript?

We observed much higher gas fluxes at FAV1, preventing mixing with air.

How does the 2014 pmoA gene library compare with the one generated here?

With the metagenome approach (see above) no library is created. The amplicon pmoA library of the 2016 paper will be biased by the primers used while the metagenome sequence data do not depend on PCR amplification.

If the authors were to more directly engage the temporal component of the data collected over the last decade, a deeper understanding of how volcanic emissions and microbial processing change with time could be developed.

We will go for the underlined option (see above) and have included more discussion (comparison) with the older work. See also remarks of Reviewer #1.

2. The quantification efforts - in both the 16S rRNA genes (Fig. 1) and the carbon and hydrogen cycling (Fig. 3) - present some problems. The 16S data would be more representative of the community structure and influence if it were scaled by the overall biomass for each horizon. As currently presented, the figure (and all aspects of the text) suggest that there was equal abundance of microbial cells at all horizons. Was this in fact the case? While cell counts would probably be the best way to derive appropriate scaling factors, such data may not be available; the quantified extracted DNA from the Qubit would be a decent back-up option. By scaling the relative abundance by the actual abundance, the authors can provide a more realistic sense of how communities at different horizons compare. For example, aspects of the "community analysis" section could change substantially; e.g., statements such as "the amount of Methylococcaceae remained more or less constant." Of greater importance, perhaps, is the quantification of functional genes and pathways; I did not find statements such as "13.1% of the retrieved genes encode for proteins involved in the CH₄ oxidation pathway" particularly useful. This is because the relevant denominator isn't well-constrained. For example, if one pathway involves 20 genes, and another involves 2, one might expect many more of the recovered genes to map to the first pathway even if the second pathway is recovered in its entirety. Gene length should also be taken into account in these assessments. In addition, the use of genes for multiple pathways (see comment on mdh below) could result in over-counting of pathway representation.

Soil cell counts are impossible in these soils. Published metagenomic studies do not involve cell counts but more refers to the amount of DNA extracted. However, still comparison between different soil types may be difficult. We have added the DNA amounts extracted to the manuscript.

Quantifications can only be relative to the amount of extracted DNA. **Relative abundance** in the total DNA extracted. Same amount of soil used for each DNA extraction This is the maximum achievable quantification.

For a longer gene you will have originally more reads. The use of marker genes and not full pathways is a very common approach in metagenomic studies.

3. Lines 479-482 indicate that separate metagenomes were acquired for each horizon: 0-10, 10-15, and 15-20cm. If this is the case, then why were the metagenomes - and all of the associated functional genes and metabolic pathways - not analyzed separately, by horizon? Given the vertically resolved geochemical and 16S analyses, there is much to be gained by disaggregating the metagenomic data. Does the relative abundance of methanotrophy, or various carbon fixation pathways, change with depth? The clear vertical redox zonation shown in Table 1 offers a compelling lens through which to view putative metabolic pathways.

DNA extraction of these type of samples is extremely difficult, e.g. no DNA at all from FAV1, The amount of reads we obtained from the different depths of FAV2 are not sufficient to reconstruct a metabolic network per layer. For this reason we had to combine the reads before analyses of marker genes for the different metabolic processes.

4. The reliance on marker genes as representatives of entire pathways seems unnecessarily limiting. In my experience, marker genes are typically used with specific primers, when metagenomes are not available. In the case of this manuscript, a metagenome should in theory allow for reconstruction of full pathways. In the discussion on methanotrophy and methylotrophy, for example, why did the analysis not include formaldehyde dehydrogenase and formate dehydrogenase? A more thorough analysis of full pathways could expose metabolic bottlenecks, potential metabolic crosstalk, and patterns in gene copy number. When normalized by genome coverage and gene length, it would also help make any quantitative treatments based on the data more credible.

Marker genes are not only used for amplicon (primer driven PCR) sequencing but also in metagenome analysis. They provide very good indication on the presence of defined metabolic activities. A 'full pathway' is difficult to identify since a lot of enzymes overlap in different metabolic pathways. As an example: pmo and xoxF/mxaFI are real marker genes for methanotrophy and methylotrophy. However, formaldehyde dehydrogenase and formate dehydrogenase also occur in non-specialized bacteria.

As stated above (3.), the amount of reads are not sufficient to reconstruct a metabolic network per layer. For this reason we combined the reads before analyses of marker genes for the different metabolic processes.

Additional Comments:

Lines 28-31: The references to "top layer," "increases with depth," and "deeper layers" are not particularly useful without reporting what those depths are. Perhaps the "various depths" mention in line 27 could include a parenthetical note on which depths were sampled for community structure.

Specification of "various depths" added.

Lines 37-39: I'm not sure the percentage values for carbon fixation pathways are useful or necessary in the abstract; what are these percentages of? It might be best to leave the percentages for the main text where the method for determining percentages can be better explained.

We agree and removed the values from the Abstract but included that the reverse TCA cycle is the most abundant.

Introduction: Additional background on the field site would help place this work in a more holistic geobiological context. For example, what type of volcano is Pantelleria, how long has it been active, and how do its emissions (content and quantity) compare with other volcanoes? Why was this site chosen for this work? How were FAV1 and FAV2 selected - why were more sites not sampled? A figure showing the field site would also be very helpful. This figure could include a map, images from the field, and labels of where precisely the two soil cores were collected.

This part has been rephrased and extended in the revised manuscript. Description of the type of volcano and references added. The field site map is available from Gagliano et al. 2016.

Added references are:

Scaillet, S., Rotolo, S.G., La Felice, S., Vita, G., 2011. High-resolution $^{40}\text{Ar}/^{39}\text{Ar}$ chronostratigraphy of the post-caldera (<20 ka) volcanic activity at Pantelleria, Sicily Strait. *Earth and Planetary Science Letters* 309, 280–290
<http://dx.doi.org/10.1016/j.epsl.2011.07.009>

Rotolo, S.G., La Felice, S., Mangalaviti, A., Landi, P., 2007. Geology and petrochemistry of the recent (<25 ka) silicic volcanism at Pantelleria island. *Bollettino della Societa Geologica Italiana* 126, 191–208.

Fulignati, P., Malfitano, G., Sbrana, A., 1997. The Pantelleria caldera geothermal system: data from the hydrothermal minerals. *J. Volcanol. Geotherm. Res.* 75, 251–270

Fiebig, J., Hofmann, S., Tassi, F., D'Alessandro, W., Vaselli, O., Woodland, A.B., 2015. Isotopic patterns of hydrothermal hydrocarbons emitted from Mediterranean volcanoes. *Chem. Geol.* 396, 152–163. <http://dx.doi.org/10.1016/j.chemgeo.2014.12.030>.

Line 66: Please provide a reference for this sentence.

Reference included.

Line 90: This reference pertains to research conducted in anoxic submarine sediments, which are not so analogous to the subaerial systems being discussed in this manuscript and implied in this paragraph. At marine mud volcanoes, methane is almost entirely consumed either in the sediment or water column, and thus are less likely than subaerial volcanic systems to "contribute to the world's CH₄ emissions."

We agree and removed the second (speculative) part of the sentence.

Lines 102-107: This list of carbon fixation pathways does not, in my opinion, add much to the introduction and could be removed. In its place, perhaps the authors could describe and explain the carbon fixation pathways typically found at geothermal environments. Do such environments shape carbon fixation approaches in ways that are substantively different from "normal" soils or less selective environments?

There is hardly any knowledge on carbon fixation pathways in volcanic ecosystems therefore we would like to keep this part on possible pathways.

Lines 114-116: Previous work at this site has largely "determine[d] the important gaseous electron donors supporting the chemolitho(auto)trophic microbial community," which is the stated goal of this manuscript. Perhaps the authors could better explain how this study goes beyond the earlier work. The focus on hydrogen could be helpful, and lines 118-120 are good, but it would be nice to see a more clear distinction from previous work in the stated goals. (If things are substantially restructured based on major comment #1 above, the stated objectives could also include a longer-term analysis of how the microbial community changes at this site.)

See also Editor remark. We have modified the last paragraph of the Introduction. We changed 'supporting' into 'that could support' and made a direct coupling to the investigation of the microbial key players.

Lines 128-130: This assertion is not well supported, and could be interpreted as contradicting the later suggestion - and foundational premise of this study - that the microbial community influences the gas content. Relevant references would be appreciated here, and a bit more clarity on the idea being put forth. Are the hydrothermal gases flowing upward from different reservoirs? Do they react

along the way in a manner that changes their composition? And if so, then aren't the differences due more to what is encountered along the way rather than simply a different path?

Geothermal gases rise up always through fractures and faults. Soils in geothermal areas are very complex systems. Cracks are present in all types of soil but in actively degassing geothermal soils like those of Pantelleria their importance is much higher because gas is often pressure driven contrarily to normal soil where gas movements are generally only driven by concentration gradients. Gas flow is therefore often focussed in small areas sometimes becoming open vents (fumaroles) where hydrothermal gases are directly released to the atmosphere. Of course these heterogeneities are reflected in variations from areas dominated by purely pressure driven gas exhalation to areas dominated only by concentration gradients. The gases themselves interact with and modify the soils, the most effective agents being temperature and pH. Furthermore, another important parameter is vapour condensation temperature and the depth at which it is reached. Condensed water is the most important agent. Due to very different solubility in water, it may influence directly the gas composition subtracting part of the most soluble ones. It may strongly increase the alteration of soil mineral constituents. It may favor the reaction of the gases with the soil depositing carbonates, sulfates, sulfides etc. It may change the permeability of the soil both on the long term by forming clay minerals and the above-mentioned secondary minerals and on the short term by favoring the swelling of all these minerals due to hydration. This study is therefore especially important, because with geochemical methods alone it is impossible to distinguish changes in gas composition due to microbial activity even if they are very important.

We have added text (and references) to support the assertion made. Adapted in the manuscript. Reference added:

Chiodini, G., Granieri, D., Avino, R., Caliro, S., Costa, A., 2005. Carbon dioxide diffuse degassing and estimation of heat release from volcanic and hydrothermal systems. *J. Geophys. Res.* 110, B08204.

Lines 131-136: References and a little more detail would be helpful here. What kinds of gas-water-rock interactions are relevant here?

Acid weathering of volcanic glass and minerals due to dissolution of CO₂ and sulfuric acid deriving from H₂S oxidation.

How exactly do depletions and enrichments happen?

During their ascent to the surface, the volcanic/geothermal gases are depleted in some species (such as SO₂) and enriched in others (such as CH₄ and H₂S) due to the strongly reducing environment in geothermal systems (Giggenbach, 1980).

How is permeability lowered - are there precipitation reactions that occur?

See previous answer.

Text modified accordingly and references added:

Giggenbach WF (1980) Geothermal gas equilibria. *Geochim Cosmochim Acta* 44:2021–2032

Lines 141-142: The production of sulfuric acid via sulfide oxidation as suggested in ref. 12 is intriguing, but I wonder if there is any evidence of this. It doesn't seem that any known sulfide oxidizers or epsilonproteobacteria were found at FAV1 previously...

We agree and deleted this part.

Lines 142-143: It is often challenging to link concentration of a metabolite to putative metabolic activity using said metabolite. For example, higher sulfide concentrations could indicate that sulfide production is simply faster than sulfide consumption, but both values could be very high.

We agree and deleted this part.

Lines 149-153: The dueling interpretations of O₂ and N₂ increases and H₂ and CH₄ decreases at the surface lack consistency. If atmospheric air is permeating the soil to increase O₂ and N₂, why wouldn't it simultaneously be diluting H₂ and CH₄? The biological activity explanation is certainly a possibility, but at this point in the discussion, I'm not sure we can rule out the abiotic scenario.

We included that we cannot rule out an abiotic scenario and extended the text. Simple air dilution cannot explain alone at least the H₂ decrease because the CO₂/H₂ ratio increases towards the surface. The CO₂/CH₄ ratio remains more or less constant or increases only slightly, but as shown in (18) the highest methanotrophic activity was in the very last cm of the soil.

Lines 154-156: The preferential decrease of H₂ is an interesting finding; in addition to the biological consumption possibility, do the authors think its smaller molecular size and higher mobility through porous media could be a relevant factor?

We do not think that this plays a role. It is theoretically possible but it could not be measured with our analytical precision. This paragraph was modified.

Lines 168-169: Some additional information on the use of two extraction methods would be useful, either here or in the methods section. What are the specific known biases of these two methods? Do the two methods complement each other in that sense, making up for the other's blind spots? If you were to disaggregate the 16S data, did the effort to avoid biases work? (This is addressed with the methanotrophs in lines 236-239 and Fig. 2, but extending such an analysis beyond those lineages would be very helpful.)

See above, we improved the description. More details about the DNA extraction methods were added in the methods.

Lines 172-173: 62 16S rRNA genes seems surprisingly low. I realize this is because they were derived from the metagenomic data and they are full length, which helps provide more taxonomic resolution, but might the authors comment on this number, especially in comparison to the 2016 paper, where FAV2 produced 147 OTUs? Is there a reason an additional high-throughput 16S-specific sequencing run was not performed?

The 2016 study is amplicon sequencing and cannot be compared to metagenome studies. The amount of 16S rRNA genes/reads is what could be expected when compared to the total amount of reads obtained.

Lines 181-183: I agree this is surprising, given the earlier measurements of methane's $\delta^{13}\text{C}$ values of -17.5 and -17.2‰ , which is far from the traditionally accepted biogenic range of values. How do the authors reconcile their detection of methanogens with the isotopic measurements from D'Alessandro et al.? What kind of mixing ratio might be required to allow for methanogenesis but retain these heavy carbon values?

The above mentioned isotopic composition of CH₄ was measured in fumarolic gases where biogenic contribution should be excluded due to temperatures $> 100^\circ\text{C}$. Unfortunately we have no isotopic data for CH₄ in the soil gas.

Lines 189-199: This section on attempts to reconcile methanogenic activity with aerobic soils is great - it succinctly explains the challenges, cites important recent studies on the subject, and proposes a possible explanation.

Thank you, we agree.

Lines 202-204: Is there a reason these data on ammonium concentrations are not shown? I think they would be very helpful in allowing the reader to compare the site to other locations and evaluate the biogeochemical impact the ammonia oxidizers could be having.

Data are included, see also above.

Line 205: This assertion seems reasonable based on figure 1, but it might be more convincing to provide alpha diversity statistics of each horizon and the archaeal and bacterial communities associated with each. This information would also help unpack just how much more diverse the bacterial community was.

We have added a Supplementary figure showing the alpha diversity between bacteria and archaea in the three layers.

Lines 214-215: This sentence is somewhat misleading: "this study" refers to the Gagliano 2016 study (right?), but could be interpreted as the Picone et al. manuscript itself.

To avoid the misleading we have re-written the sentence.

Lines 217-222: This section is interesting, and the new detection of a more substantial verrucomicrobial component represents an important result in comparison to past analyses. Does the PowerSoil kit (which I believe was used in Gagliano et al., 2016?) specifically miss Verrucomicrobia?

We think it is not the DNA extraction method but more likely the choice of the primers used to amplify (parts of) the 18S rRNA genes.

Lines 232-234: A brief account of the gammaproteobacterial aerobic methanotrophs' acid tolerance would be welcome here, in order to bolster the point that pH may be a selective factor. Islam et al., Frontiers in Microbiology, 2016 could be a good place to start.

We have included a brief account on acid tolerance.

Lines 274-277: The specific analysis of pmoA vs. amoA is very helpful, and the lack of amoA sequences is unexpected given the presence of the Thaumarchaea. How do the authors explain this discrepancy? Could the relative coverage of the thaumarchaeal genomes help? For example, if the genome is only 5% closed, one might not expect to find the amoA gene. Comparing coverage of the methanotrophs with the Thaumarchaea - as well as the gene copy number and lengths of their respective functional genes - could be interesting.

If present the gene-driven approach would have identified possible amoA sequences. Also the bins were checked and no evidence for nitrification was found. It should be kept in mind that not all Thaumarchaea are ammonia oxidizers/

Lines 281-284 and throughout: For all of the functional genes, it would be nice to know how many of them mapped onto contigs that also had a 16S rRNA gene. In other words, how many of these taxonomic inferences of functional genes are relatively certain, and how many are circumstantial?

In the gene oriented approach it is the presence of functional gene in a contig. No coupling to taxonomy is aimed at in first instance. What we did is combining the hydrogenase, pmo, mdh marker genes to their phylogenetic origin. This is presented in the phylogenetic trees. Regularly this nicely reflects the taxonomic position of the microorganism carrying the gene.

Figure 4: Perhaps, as with figure 2, the taxonomic assignments could be placed alongside the vertical colored bars on the right; this saves the reader a step, and eliminates the need for a legend.

We changed the figures as requested.

MDH discussion: Since MDHs can be used by non-methanotrophs such as methanol oxidizers, how do we know the 46 recovered mdh genes are only being used for methanotrophy?

We do not state that all the retrieved MDHs are only used for methanotrophy. For example, we specify that *XoxF3* is found in N_2 fixing microorganisms, methanotrophs and methylotrophs. However, to avoid confusion we have added a separate sentence at the start of this paragraph.

Table 3: What could the offset between the number of genes and the relative abundance of genes be indicating? Do any of these hydrogenases map back onto a 16S rRNA gene that could help establish a link between gene : relative abundance ratio and relative abundance of the associated organism?

In the gene oriented approach it is the presence of functional gene in a contig. No coupling to taxonomy is aimed at in first instance. Linking 16S rRNA genes to hydrogenases is challenging. Many different micro-organisms have hydrogenases and the role of hydrogenases can be different. Autotrophic growth on H_2 cannot be as easily detected with 16S genes as is the case for methanotrophs.

Lines 355-357: Here, as well as with the discussion of methanogens in aerated soils and throughout the manuscript, it might be helpful to consider that the presence and recovery of genes does not mean that the associated enzyme or pathway are active. Could it be possible that these genes are left over from a time when conditions were anoxic? Or that the associated organisms were transported from anoxic zones? Is the gas emission strong enough to transport organisms from depth into the upper reaches of the soil column?

We have included the arguments mentioned by the reviewer in the discussion of methanogens.

Lines 358-362: Listing other sites of hydrogenase recoveries is not particularly informative without an interpretation. For example, what is it about "forest soil and permafrost" - the microbial communities, the environmental parameters - that could explain why organisms with NiFe-hydrogenases are found both there and in the Favara Grande samples?

Greening et al., 2016 (25) suggest that oxygen partial pressure is the principal driving force for the distribution of hydrogenase in a specific environment. Additionally, pH, temperature and metal ion availability may play a role. They also state "However, experimental studies are required to gain a deeper understanding of the ecological significance of H_2 oxidation and evolution". We have extended this paragraph mentioning the environmental parameters.

Lines 364-392: This section strikes me as a list of carbon fixation pathways that could benefit from additional discussion and interpretation. For example, lines 372-375 is so high-level as to offer very little insight. I'm not sure exactly what the best way to enhance this section would be, but two options could include the following. 1) Getting more granular to see how specific lineages process carbon and could interact directly or indirectly with other organisms. 2) Getting even higher-level to see how this distribution of carbon fixation pathways within a given microbial ecosystem compares with other types of systems. And what would the diversity of carbon fixation approaches tell you about the system and its redox zonation, energy availability, available niches?

We agree that this part gives a very general overview of carbon fixation pathways present. However, we would like to keep the text as it is since additions would result in too much speculation.

Lines 395-400: I'm not sure I understand the reasoning here. If there "is probably not much of a niche for ammonia oxidizers," then how were so many recovered via 16S rRNA analysis? Perhaps a different interpretive track to pursue is the possibility of a low standing stock of ammonium - that it may be produced or introduced, but that it's consumed very quickly.

Not all Thaumarchaeota are capable of ammonia oxidation and employ different metabolic strategies to survive. We have modified the text and added a reference.

Lines 403-404: Why was nitrogenase activity "most likely involved in N₂-fixation for biomass production"?

Nitrogenase is the crucial (and only) enzyme for nitrogen fixation which is a very energy costly process. Little nitrogen is available in this ecosystem, but N₂ is available.

Discussion / Conclusion: What are some concrete next steps this research could take to address some of the (many) remaining questions raised in this manuscript? For example, the source of the methane, the ammonium paradox, the DNA recoverability issues, etc.

We have included additional concrete steps.

Line 417: Perhaps "emissions" should be replaced by "concentrations," as no flux measurements are reported in this manuscript.

Changed accordingly.

Lines 427-428: Presumably you mean they "have not been detected" at this site before?

Changed accordingly.

Lines 438-441: What was the reasoning behind the selection of sampling depths in this study? Why 11 cm and not 10, and why 30 and 50 but not 40? Why not deeper, or more compressed (e.g., 5 cm) horizons?

We considered 11 cm with our sampling system the shallowest depth allowing a gas sampling with no direct air contamination, while 50 cm was considered the depth where we should have found the almost pure hydrothermal component.

Lines 439-440: A little more detail on the use of pH strips might be helpful. What is the margin for error with these? If it's what I'm envisioning, you match the color to the closest 0.5 pH unit, which suggests that if the closest color is 4.0, then the actual pH is between 3.75-4.25? Were the user's visual approximations calibrated in the lab with a pH probe? Doing so would help establish the reported measurements as more trustworthy.

These pH strips give a good indication of the pH value. Giving the remote location of the sampling site, bringing a pH electrode in the field was not feasible.

Lines 441-443: I find a couple of items in this description slightly confusing. First, when collecting 60 mL of gas, can you calculate the expected "sphere of influence" from which this gas was drawn? If we presume equal sucking power in all directions, this can be calculated by measuring the porosity at each horizon. This would give a range of depths that would more accurately represent the situation; the "30 cm" horizon might in fact be, 28-32 cm or something.

This would be true if no pressure gradient would be present in the soil. With our very low suction speed we try not to exceed the natural gas flux. When no pressure gradient is present it is anyway very difficult to calculate a sphere of influence because soils are highly anisotropic and it is very difficult to obtain reasonable average "connected" porosity values.

Second, what kind of gas were the sampling vials flushed with? If it was gas from the horizon of interest, it seems, as written, that 150 mL of gas were used for this...and yet only 60 mL were collected? Please clarify both of these points.

The volume of sampled gas corresponds to the volume of the sampling vial (12 ml) used to collect the sample. Due to the fact that the vial is initially filled with air, it is common practice to consider the sample uncontaminated if the vial has been flushed with a volume of gas 10 times greater than the vial itself. Therefore the vial was flushed with about 150 ml of soil gas.

Lines 445-446: Why was the top cm of soil removed from the analysis, and how do the authors imagine this changed their results? I would expect a distinct, photosynthetically driven community at the air-soil interface - omitting this horizon from the analysis likely changed the diversity results substantially.

The first 1 cm is probably very different and very close to atmospheric air and possible disturbance, Therefore this part was removed.

Line 452: To clarify, were gas and DNA sampled collected from the same core? If so, how was contamination from inserting the capillary tube at different depths ruled out? If not, this should be mentioned, and the potential repercussions should be addressed. For example, given the heterogeneity found between FAV1 and FAV2 ("only a few meters apart") how would taking parallel cores change the community structure and metabolic interpretations? (Incidentally, Gagliano et al., 2016 described FAV1 and FAV2 as being "about 10 m apart;" please explain this inconsistency.)

Of course we didn't collected the soil and the gas sample exactly at the same place but about 10 cm apart. At this distance we should be reasonably sure not to have significant differences. Anyway there was no other way to operate and 10 cm are much less than ten meter. The location was identical to Gagliano et al (2016). So we changed "only a few meters apart" into "about 10 meters apart".

Line 466 and throughout: I found it slightly distracting that the horizons were labeled inconsistently as "top layer," "10-15 cm," and "15-20 cm." It might be easiest to indicate what the top layer depth range was. Earlier I believe it was specified as 0-10 (line 167)...which, in reference to the comment above, suggests the geochem and molecular work was done on separate cores?

Changed accordingly in text and figure.

Lines 465-468: The failure to recover DNA from FAV1 is perplexing, and additional troubleshooting efforts or explanations would be welcome. Previous work was able to recover sequenceable DNA from this site; why was this effort less successful, despite using two extraction methods and, it appears, having more material to work with (10 cm horizons vs. 0-3 in Gagliano et al., 2016)? To what do the authors attribute the poor extractions? Soil chemistry, lack of biomass, etc.? Were cell counts conducted to double-check the relative amounts of biomass? I would also be curious to know what the DNA quantification results showed, and believe this is important for overall interpretation of the results (see major comment 2 above).

The previous work did not recover sequenceable DNA since they used the DNA for amplicon PCR sequencing. For this amplicon sequencing approaches you only need very little amounts of DNA. Our yield was not enough to do Illumina metagenome sequencing. In addition the quality of the DNA was very bad and purification attempts resulted in loss of all DNA. The problems of extracting DNA from acid soils is reported more often in literature and we know it from own experience.

Lines 497-499: I am admittedly not a metagenomics expert, but this seems like a lot of binning algorithms! Why were all of them necessary? Did they all agree with each other? How was data from one algorithm used to seed the next algorithm? Further explanation of why so many approaches were used - and why each one was necessary - would be appreciated.

It is not unusual to combine different algorithms for binning. The individual results are as indicated used for consensus binning with DAS Tool 1.0. The reference cited for DAS Tool 1.0; Sieber et al. 2018 provides a good explanation.

Lines 501-504: More detail on the metabolic reconstruction approach is needed to allow the reader to fully understand and evaluate what happened. What is involved in the "in-house" pipeline? How were the customized HMM profiles developed and validated? What settings were used for Prodigal

and PROKKA? (In the previous section, what were the specific settings for all of the assembly and binning algorithms?)

We now supply more detail on the metabolic reconstruction and included a list of marker genes in the Supplementary information.

October 14, 2020

Prof. Huub J.M. Op den Camp
Radboud University Nijmegen
Department of Microbiology
Faculty of Science
Heyendaalseweg 135
Nijmegen NL-6525 AJ
Netherlands

Re: mSystems00517-20R1 (Geothermal gases shape the microbial community of the volcanic soil of Pantelleria, Italy)

Dear Prof. Huub J.M. Op den Camp:

Your manuscript has been accepted, and I am forwarding it to the ASM Journals Department for publication. For your reference, ASM Journals' address is given below. Before it can be scheduled for publication, your manuscript will be checked by the mSystems senior production editor, Ellie Ghatineh, to make sure that all elements meet the technical requirements for publication. She will contact you if anything needs to be revised before copyediting and production can begin. Otherwise, you will be notified when your proofs are ready to be viewed.

Sincerely,

Karen Lloyd
Editor, mSystems

Journals Department
Supplemental Material: Accept
Supplemental Material: Accept
Supplemental Material: Accept
Supplemental Material: Accept
Supplemental Material: Accept